# Genome of *Pythium myriotylum* Uncovers an Extensive Arsenal of Virulence-Related Genes among the Broad-Host-Range Necrotrophic *Pythium* Plant Pathogens

Paul Daly,[a] Dongmei Zhou,[a] Danyu Shen,[b] Yifan Chen,[a,c] Taiqiang Xue,[a] Siqiao Chen,[a,d] Qimeng Zhang,[a] Jinfeng Zhang,[a] Jamie McGowan,[e] Feng Cai,[d,f] Guan Pang,[d] Nan Wang,[a] Taha Majid Mahmood Sheikh,[a] Sheng Deng,[a] Jingjing Li,[a] Hüseyin Okan Soykam,[g] Irem Kara,[g] David A. Fitzpatrick,[e] Irina S. Druzhinina,[d,h] Günseli Bayram Akcapinar,[i] Lihui Wei[a,c]

aKey Lab of Food Quality and Safety of Jiangsu Province—State Key Laboratory Breeding Base, Institute of Plant Protection, Jiangsu Academy of Agricultural Sciences, Nanjing, China
bCollege of Plant Protection, Nanjing Agricultural University, Nanjing, China
cSchool of Environment and Safety Engineering, Jiangsu University, Zhenjiang, China
dJiangsu Provincial Key Lab of Organic Solid Waste Utilization, Fungal Genomics Laboratory (FungiG), Nanjing Agricultural University, Nanjing, China
eGenome Evolution Laboratory, Maynooth University, Maynooth, Ireland
fSchool of Ecology, Sun Yat-sen University, Shenzhen, China
gDepartment of Biostatistics and Bioinformatics, Institute of Health Sciences, Acibadem Mehmet Ali Aydinlar University, Istanbul, Turkey
hDepartment of Accelerated Taxonomy, The Royal Botanic Gardens Kew, London, United Kingdom
iDepartment of Medical Biotechnology, Institute of Health Sciences, Acibadem Mehmet Ali Aydinlar University, Istanbul, Turkey

Paul Daly and Dongmei Zhou contributed equally to this article. Author order was determined alphabetically.

**ABSTRACT** The *Pythium* (Peronosporales, Oomycota) genus includes devastating plant pathogens that cause widespread diseases and severe crop losses. Here, we have uncovered a far greater arsenal of virulence factor-related genes in the necrotrophic *Pythium myriotylum* than in other *Pythium* plant pathogens. The genome of a plant-virulent *P. myriotylum* strain (~70 Mb and 19,878 genes) isolated from a diseased rhizome of ginger (*Zingiber officinale*) encodes the largest repertoire of putative effectors, proteases, and plant cell wall-degrading enzymes (PCWDEs) among the studied species. *P. myriotylum* has twice as many predicted secreted proteins than any other *Pythium* plant pathogen. Arrays of tandem duplications appear to be a key factor of the enrichment of the virulence factor-related genes in *P. myriotylum*. The transcriptomic analysis performed on two *P. myriotylum* isolates infecting ginger leaves showed that proteases were a major part of the upregulated genes along with PCWDEs, Nep1-like proteins (NLPs), and elicitin-like proteins. A subset of *P. myriotylum* NLPs were analyzed and found to have necrosis-inducing ability from agroinfiltration of tobacco (*Nicotiana benthamiana*) leaves. One of the heterologously produced infection-upregulated putative cutinases found in a tandem array showed esterase activity with preferences for longer-chain-length substrates and neutral to alkaline pH levels. Our results allow the development of science-based targets for the management of *P. myriotylum*-caused disease, as insights from the genome and transcriptome show that gene expansion of virulence factor-related genes play a bigger role in the plant parasitism of *Pythium* spp. than previously thought.

**IMPORTANCE** *Pythium* species are oomycetes, an evolutionarily distinct group of filamentous fungus-like stramenopiles. The *Pythium* genus includes several pathogens of important crop species, e.g., the spice ginger. Analysis of our genome from the plant pathogen *Pythium myriotylum* uncovered a far larger arsenal of virulence factor-related genes than found in other *Pythium* plant pathogens, and these genes contribute to the infection of the plant host. The increase in the number of virulence factor-related genes appears to have occurred through the mechanism of tandem gene duplication events. Genes from particular virulence factor-related categories that

Address correspondence to Lihui Wei, weilihui@jaas.ac.cn, or Paul Daly, paul.daly@jaas.ac.cn.

The authors declare no conflict of interest.

were increased in number and switched on during infection of ginger leaves had their activities tested. These genes have toxic activities toward plant cells or activities to hydrolyze polymeric components of the plant. The research suggests targets to better manage diseases caused by *P. myriotylum* and prompts renewed attention to the genomics of *Pythium* plant pathogens.

**KEYWORDS** *Pythium myriotylum*, ginger, genome dynamics, virulence factors

*P*ythium species are devastating plant pathogens that cause widespread disease; e.g., *Pythium* root rot has been referred to as the "common cold" of wheat (1). The *Pythium* genus has approximately 300 species (2). *Pythium* species can be commonly found in soil and water, and *Pythium* plant pathogens tend to have broad host ranges (3). The *Pythium* genus includes plant pathogens, animal pathogens, and mycoparasites (e.g., *P. oligandrum*) as well as parasites of other *Pythium* species. In spite of the importance of the genus, only a limited number of *Pythium* species genomes have been sequenced and annotated.

The majority of *Pythium* species whose genomes have been sequenced and analyzed to date are plant pathogens (*P. aphanidermatum*, *P. arrhenomanes*, *P. irregulare*, *P. iwayamai*, *P. splendens*, and *P. ultimum*) or insect (*P. guiyangense*) or animal (*P. insidiosum*) pathogens, as well as adelphoparasites suitable for biocontrol (*P. oligandrum*). Previously, in a subset of these *Pythium* species, virulence factors such as plant cell wall-degrading enzymes (PCWDEs) (4) and putative effectors (5) have been compared at the genome level. Analysis of the expression of these virulence factors during infection by *Pythium* plant pathogens is limited, even though this is a critical factor in understanding the pathogenicity of broad-host-range pathogens, given that these pathogens likely evolved to express a subset of these effectors for infection of particular hosts. The sequencing of genomes as well as transcriptomic and proteomic analyses of oomycetes has dramatically expanded the understanding of oomycete biology (see reference 6 for a recent review), but this has mainly been in *Phytophthora* species and not in *Pythium* spp.

With regard to oomycete genome dynamics, previous analysis of 10 stramenopile genomes, including six oomycete genomes, showed that pathogenic oomycetes have gained and lost genes with gene gains through duplications greater than the gene losses (7). Recently, it was shown in the biocontrol *Pythium* species *P. oligandrum* that tandem duplication events contributed to the expansion of particular CAZy gene families (8), and previously, tandem duplications were shown to have a role in the expansion of putative effector gene families in *Phytophthora infestans* (9).

*Pythium myriotylum* Drechsler is a broad-host-range pathogen that has been shown to infect a range of plant species (10), including dicots such as *Nicotiana tabacum,* for example (11), and monocots such as ginger (*Zingiber officinale*). We have shown recently that *P. myriotylum* is a major cause of *Pythium* soft rot (PSR) of ginger in China (12). Ginger is an important crop whose rhizome is used as a spice as well as for alternative medicine (13). Ginger is widely grown in the tropics and subtropics, and global production is estimated at approximately 3 million metric tons (14, 15). Diseases of ginger rhizomes are of global interest, as the ginger rhizomes from one region can be transported to another as a global commodity. Root rot of tobacco seedlings caused by *P. myriotylum* is also a major problem for the tobacco industry (16). There are several studies that examine the global transcriptional response of ginger to *P. myriotylum*, for example (17), but none that examine the transcriptional response of *P. myriotylum* during infection of ginger. Here, we use ginger as a model host for *P. myriotylum* to understand how its gene expression patterns are altered during infection.

Our objectives were to understand the pathogenicity of the broad-host-range pathogen *P. myriotylum* by using comparative genomics with other *Pythium* species, transcriptomics, and analysis of the activities of putative virulence factor-related genes. The expansion of virulence factor-related genes in *P. myriotylum* points to its success as a broad-host-range pathogen and resets the upper limit as to the number of

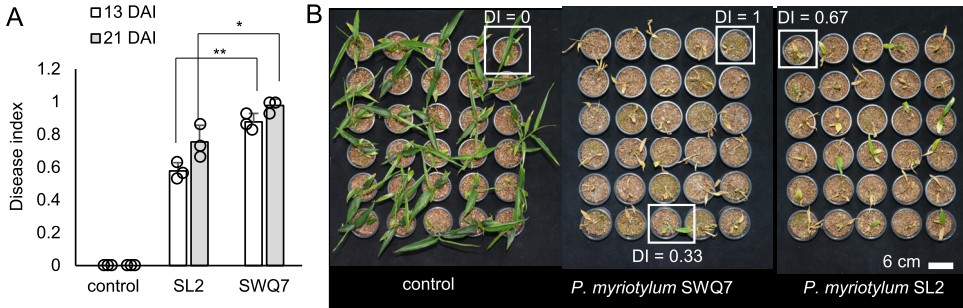

**FIG 1** Different virulence levels of *P. myriotylum* isolates toward ginger. (A) Disease index (DI) from infection of ginger plants with the *P. myriotylum* SWQ7 and SL2 isolates 13 and 21 days postinfection (dpi). *, $P < 0.05$; **, $P < 0.01$; from Student's *t* test ($n = 3$). The disease index was calculated separately for the plants on each of the three replicate sets (indicated by the empty circles on the graph) on separate shelves in the growth chamber (each shelf contained 10 control, 10 SWQ7-infected, and 10 SL2-infected plants). (B) Images of all the ginger plants arranged by treatment group on the final day disease symptoms were recorded, 21 dpi. Representative examples of the DI level corresponding to different levels of disease severity as described in Material and Methods are shown in white boxes.

virulence factor-related genes that can be found within a *Pythium* plant pathogen. Here, we uncovered the genome of *P. myriotylum* and found that it contained the most extensive arsenal of pathogenesis genes among *Pythium* plant pathogens and investigated the activities of the genes in these unique and expanded gene families.

## RESULTS

**Different virulence levels of *P. myriotylum* isolates toward ginger.** Our previous study indicated that two *P. myriotylum* isolates, SWQ7 and SL2, had relatively high and low virulence toward ginger, respectively (12). Here, the two *P. myriotylum* isolates were analyzed in a pot trial with more replicate plants and to measure differences at more time points in the virulence of the isolates (Fig. 1). These isolates are from infected ginger rhizomes from locations 100 km apart in Shandong province, one of the major ginger-growing regions in China. Here, both of the *P. myriotylum* isolates could clearly infect the ginger plants, with a significantly higher disease index (DI) from SWQ7 than from SL2 ($P < 0.05$ by Student's *t* test). The difference in disease index between the two isolates was greater at earlier than at later time points after inoculation with the pathogens (Fig. 1A). The above-ground disease symptoms progressed from initial yellowing of the lower leaves to yellowing of the remaining leaves and shoots and then death of the ginger plant. We chose SWQ7, the more virulent of the two isolates, for third-generation genome sequencing, and both of the isolates for transcriptomic analysis during infection of ginger leaves.

**High-quality genome assembly of *P. myriotylum*.** The genome of *P. myriotylum* isolate SWQ7 was sequenced using a combination of short Illumina reads and long PacBio reads. Approximately 7 Gb of Illumina 150-bp paired-end reads and 17 Gb of PacBio data were generated, with a subread $N_{50}$ length of 21 kb. Oomycete species are diploid throughout most of their life cycle, and long-read sequencing technology facilitates the assembly of diploid genomes, so an assembly strategy was required to result in a haploid assembly of the *P. myriotylum* genome to facilitate comparison with the haploid genome assemblies of other *Pythium* species. Two key tools to predict and monitor progress in generating a haploid genome assembly are k-mer-based tools (GenomeScope and Merqury) and Benchmarking Universal Single-Copy Orthologs (BUSCO), which can be used to indicate the level of haplotypic duplication.

The haploid genome size of *P. myriotylum* isolate SWQ7 was estimated to be 67.5 Mb based on k-mer analysis of Illumina short reads using GenomeScope (see Fig. S1A in the supplemental material). As expected, the two peaks in the GenomeScope spectral plot indicated that a diploid genome had been sequenced (Fig. S1A). *De novo* genome assembly of PacBio reads was performed using the Canu assembler, and the initial genome assembly was significantly larger than expected at 154 Mb compared to the 67.5 Mb

**TABLE 1** Overview of *P. myriotylum* SWQ7 genome statistics and comparison with other *Pythium* genomes

| Host and isolate | Assembly size (Mb) | No. of contigs/scaffold | $N_{50}$ scaffold length (kb) | % Heterozygosity | No. of genes | Avg gene length (bp) | Gene density (genes/Mb) | Reference |
|---|---|---|---|---|---|---|---|---|
| Plant host | | | | | | | | |
| *P. myriotylum* SWQ7 | 69.2 | 185 | 1,600 | 1.8 | 19,878 | 1,720 | 287 | This study |
| *P. aphanidermatum* DAOM BR444 | 35.9 | 5,667 | 37.4 | 3.9 | 12,305 | 1,470 | 343 | 5 |
| *P. arrhenomanes* ATCC 12531 | 44.7 | 10,978 | 9.8 | 2.8 | 13,805 | 1,339 | 309 | 5 |
| *P. irregulare* DAOM BR486 | 42.9 | 5,887 | 23.2 | 0.53 | 13,804 | 1,495 | 322 | 5 |
| *P. iwayamai* DAOM 242034 | 43.3 | 11,542 | 11.0 | 0.34 | 14,875 | 1,325 | 344 | 5 |
| *P. splendens* | 53.3 | 198 | 342.1 | 3.1 | 17,350 | 1,859 | 326 | 75 |
| *P. ultimum* var. *ultimum* DAOM BR144 | 42.8 | 975 | 773 | 1.8 | 15,297 | 1,299 | 357 | 42 |
| Fungus and oomycete host | | | | | | | | |
| *P. oligandrum* ATCC 38472 | 41.9 | 180 | 1,300 | No short reads generated | 15,007 | 1,623 | 358 | 74 |
| Animal host | | | | | | | | |
| *P. guiyangense* Su | 110.1 | 239 | 1,009 | 1.42 | 30,943 | 1,658 | 281 | 18 |
| *P. insidiosum* Pi-S | 53.2 | 1,192 | 146 | No short reads available | 14,850 | 1.498 | 281 | 73 |

estimated for the haploid genome from the k-mer analysis of the short reads. Analysis using BUSCO identified a large number of duplicated BUSCO genes (87.6%), indicating the presence of haplotypic duplications. The k-mer spectrum analysis of the initial genome assembly using Merqury showed a typical k-mer spectrum for a heterozygous diploid genome assembly and revealed a substantial level of haplotypic duplication indicated by substantial proportions of two-copy k-mers (blue color in plot) comparable to the number of 1-copy k-mers (red color in plot) (Fig. S1B). Haplotypic duplication was identified and removed using the purge_dups pipeline, and the assembly was polished with Illumina short reads using Pilon. The final assembly was 69.2 Mb (Table 1), which is in line with the estimated genome size of 67.5 Mb from the k-mer analysis of Illumina short reads. The k-mer spectrum analysis using Merqury showed a large reduction in the number of two-copy k-mers (blue color in plot), indicating the haplotypic regions had been removed (Fig. S1B). Unlike in the previous BUSCO analysis, now only 3.4% of BUSCO genes were reported as being duplicated, and the assembly was also highly complete, with a BUSCO completeness score of 95.7% (Table S1). The assembly is highly contiguous with 185 contigs and an $N_{50}$ of 1.6 Mb (Table 1). The consensus quality value (QV) was estimated to be 39 by Merqury and is a high quality and close to a 99.99% accuracy of the consensus assembly. The assembly size was 30% to 100% larger than most of the other *Pythium* genomes, except for *P. guiyangense*, whose large genome size is considered to be due to a hybridization event (Table 1). Also, the predicted assembly size of the *P. myriotylum* SWQ7 isolate based on k-mer analysis of Illumina short reads was similar to that of another *P. myriotylum* isolate, SL2 (data not shown).

A phylogenomic analysis was carried out to confirm the evolutionary relationship between *P. myriotylum* and other sequenced *Pythium* species. Two *Phytophthora* species were included as outgroups. Maximum likelihood analysis was performed on a concatenated supermatrix alignment of 145 BUSCO proteins that were present as single copies in at least 11 of the 12 species. The *P. myriotylum* SWQ7 genome clustered most closely with *P. arrhenomanes* and *P. aphanidermatum* with full bootstrap support (Fig. 2), which is consistent with previous phylogenetic and taxonomic analyses (3). The phylogeny in Fig. 2 also highlights the other *Pythium* spp. with sequenced genomes that were used for comparative genomics analyses with *P. myriotylum* and the differing

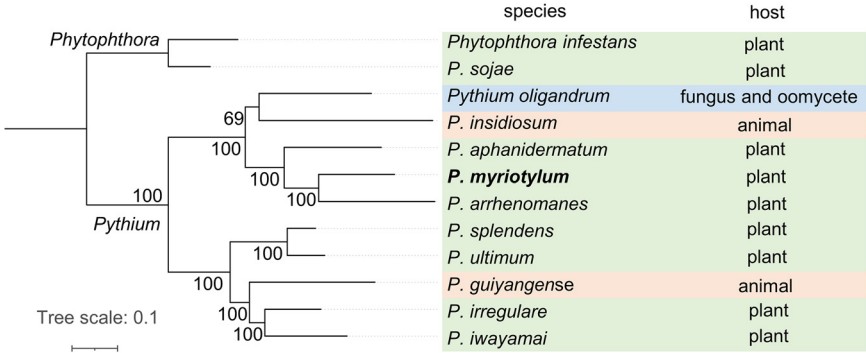

**FIG 2** Maximum likelihood-based phylogenomic analysis of *P. myriotylum* SWQ7 with other sequenced *Pythium* genomes. Here, 145 proteins that are present as single copies in at least 11 of the 12 species were used. *P. infestans* and *P. sojae* were used as outgroup species. The tree also highlights the different hosts of the *Pythium* species.

hosts of these pathogenic species. The majority of *Pythium* spp. with sequenced genomes are plant pathogens, but there is also a fungal and oomycete parasite, *P. oligandrum*, and two animal pathogens, *P. guiyangense* (insect) and *P. insidiosum*.

The $N_{50}$ scaffold length of the *P. myriotylum* assembly was greater than that of any of the other *Pythium* genomes sequenced using long-read sequencing technology, indicating a high quality of the assembly (Table 1). The heterozygosity rate (estimated from short sequencing reads using the GenomeScope tool) for *P. myriotylum* (1.8%) was intermediate to that of the other *Pythium* genomes, with the highest heterozygosity rate in *P. aphanidermatum* (3.9%) and the lowest in *P. iwayamai* (0.34%) (Table 1). The total number of predicted genes of 19,878 was 15% to 62% greater than that of most of the other *Pythium* genomes, with the exception again being the hybrid *P. guiyangense* (Table 1). The larger total number of genes in *P. myriotylum* does not positively correlate with the larger genome size, because the gene density of 287 genes per Mb for *P. myriotylum* was lower than the gene density in the genomes of other *Pythium* plant pathogens and of *P. oligandrum* and similar to that of *P. guiyangense* and *P. insidiosum* (Table 1).

In *P. myriotylum*, there was a relatively high interspersed repeat content of 31% (Table S4). Long interspersed nuclear elements (LINEs) and long terminal repeats (LTRs) accounted for ~4.5% and ~15% of the *P. myriotylum* genome sequence, respectively. The *P. myriotylum* genome repeat content is relatively high compared to what has been reported for other *Pythium* species; for example, for *P. guiyangense*, a repeat content of 6% was reported (18). Direct comparisons of repeat content with genomes sequenced by second-generation sequencing technology are complicated by the underestimation of the repeat content from short-read sequencing. To remove transposable elements from the *P. myriotylum* genome annotation, we used a protein-based repeat masking strategy after gene model annotation by filtering out proteins that matched the TransposonPSI library. This approach removed 2,779 gene models, leaving 19,878 gene models in the transposon-filtered gene model set.

**Major expansion of gene families in *P. myriotylum* compared to those of other *Pythium* species.** The 19,878 genes in the *P. myriotylum* genome were analyzed to determine which had orthologs in all other *Pythium* genomes (core) or in a subset of these genomes (dispensable) or were unique to *P. myriotylum*. Of the *P. myriotylum* genes, 14,868 were categorized as core, 3,740 genes were in the dispensable category, and 1,268 genes were unique to *P. myriotylum* (Table S2). Two notable Pfam domains exclusive to these unique genes (i.e., Pfam domains not found in the *P. myriotylum* core or dispensable genes) were PF03664 for $\alpha$-l-arabinofuranosidase and PF03068 for protein-arginine deiminase activities. To gain an overview of the functional differences between the *P. myriotylum* genes and those of other *Pythium* species, enrichment analysis of the gene ontology (GO) terms annotated for *P. myriotylum* genes was used. For the analysis of enrichment of GO terms, the ratio of a particular GO term to the total annotated GO terms in *P. myriotylum* was compared to the same ratio in three sets of

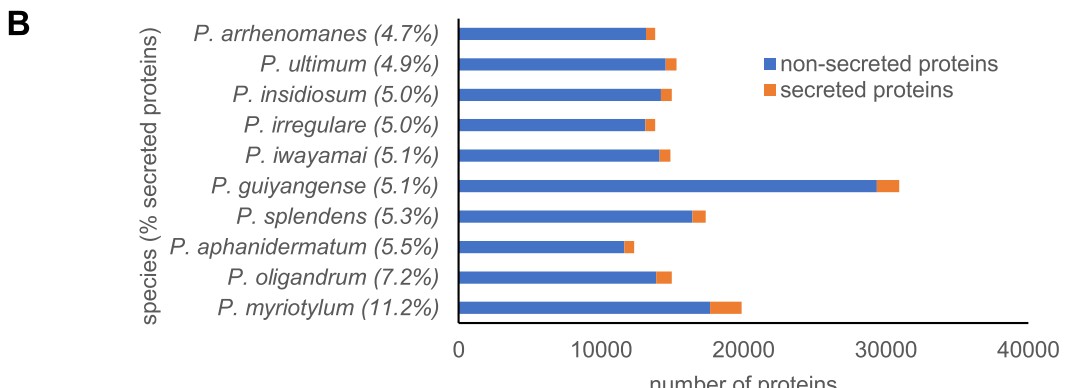

**A**

| Gene ontology (GO) term number and name | | ratio in *P. myriotylum* | ratio in all *Pythium* spp. | p value | ratio in *Pythium* plant pathogen spp. | p value | ratio in *Pythium* non-plant pathogen spp. (and *P. myriotylum*)* | p value |
|---|---|---|---|---|---|---|---|---|
| GO:0016787 | hydrolase activity | 1,577/7,131 | 11,518/64,459 | <0.00 | 7,692/40,603 | <0.00 | 5,403/30,987 | <0.00 |
| GO:0004553 | hydrolase activity, hydrolyzing O-glycosyl compound | 203/7,131 | 1,045/64,459 | <0.00 | 721/40,603 | <0.00 | 527/30,987 | <0.00 |
| GO:0006508 | proteolysis | 517/7,131 | 3,265/64,459 | <0.00 | 2,103/40,603 | <0.00 | 1,679/30,987 | <0.00 |
| GO:0008233 | peptidase activity | 532/7,131 | 3,357/64,459 | <0.00 | 2,161/40,603 | <0.00 | 1,728/30,987 | <0.00 |
| GO:0004650 | polygalacturonase activity | 27/7,131 | 53/64,459 | <0.00 | 49/40,603 | <0.00 | 31/30,987 | <0.00 |

Species columns above: *P. aphanidermatum, P. arrhenomanes, P. guiyangense, P. insidiosum, P. irregulare, P. iwayamai, P. myriotylum, P. oligandrum, P. splendens, P. ultimum* (all *Pythium* spp.); *P. aphanidermatum, P. arrhenomanes, P. irregulare, P. iwayamai, P. myriotylum, P. splendens, P. ultimum* (plant pathogen spp.); *P. guiyangense, P. insidiosum, (P. myriotylum), P. oligandrum* (non-plant pathogen spp.)

*note that *P. myriotylum* GO terms are also included in this ratio as part of the total or background calculation for the enrichment statistic

**FIG 3** Enrichment for secreted activities in the *P. myriotylum* genome. (A) Comparison of gene ontology (GO) terms enriched in the *P. myriotylum* genome compared to those of other *Pythium* spp. (B) Number and percentage of proteins from *P. myriotylum* that are predicted to be secreted and comparison with other *Pythium* spp.

the *Pythium* species and a Bonferroni correction was applied to the GO enrichment analysis *P* values, i.e., all the *Pythium* species with sequenced genomes, all the *Pythium* plant pathogens, and the non-plant-pathogenic *Pythium* species (*P. guiyangense*, *P. insidiosum*, and *P. oligandrum*). The GO term for hydrolase activity (GO:0016787) was enriched in *P. myriotylum* compared to all *Pythium* species (*P* < 0.01), as well as compared to the non-plant-pathogenic *Pythium* species (*P* < 0.01) and the *Pythium* plant pathogens (Fig. 3A and Table S3). GO terms for two broad groups of hydrolase activities (GO:0004553 and GO:0006508) were enriched in *P. myriotylum* compared to other *Pythium* plant pathogens, suggesting a prominent role for these hydrolase activities in the pathogenicity of *P. myriotylum*. These hydrolyase activities were those active on O-glycosyl compounds, found for example in polysaccharides, and those active on peptide linkages in proteins (Fig. 3A). This enrichment for these hydrolase activities, which are often secreted (99 and 382 of the proteins annotated with GO:0004553 and GO:0006508, respectively, were predicted to be secreted), correlated well with *P. myriotylum* having the highest percentage of secreted proteins (11.2%) compared to all the other *Pythium* species and at more than twice the percentage of every other *Pythium* plant pathogen (~5%) (Fig. 3B).

The total number of putative PCWDEs in *P. myriotylum* (274 genes) (Table 2) and the total number of other virulence factor-related genes in *P. myriotylum* (602 genes) (Table 3) is approximately two to three times higher than that in other *Pythium* plant pathogens. PCWDEs were annotated based on the presence of CAZy domains and putative activities associated with those domains, and other virulence factor-related genes were annotated primarily based on Pfam domains. PCWDEs can contribute to the virulence of plant pathogens by facilitating access of the pathogen to the host and contributing to the death of the host plant as well as facilitating feeding on the dead host. PCWDEs are particularly important for the lifestyle of necrotrophic pathogens such as *P. myriotylum*. Strikingly, *P. myriotylum* had the largest number of PCWDEs in comparison to that of the other *Pythium* spp. (Table 2 and Table S6). The number of cellulases is ~2- to 4-fold higher than that of most other *Pythium* species, with the exceptions of *P. splendens* and *P. guiyangense*, which have numbers of cellulases more similar to that of *P. myriotylum*. *P. myriotylum* contains far more endoxylanases (from CAZy families GH10 and GH11) than any other *Pythium* species, with *P. myriotylum* containing 11 endoxylanases and most of the other *Pythium* species containing no endoxylanases, with the exceptions of *P. aphanidermatum* and *P. arrhenomanes* (Table 2). Correlating well with its having the largest number of xylanases, *P. myriotylum* was the only *Pythium* species containing genes encoding GH62 family members in the CAZy analysis. Furthermore, in the ortholog analysis, all four of the GH62 family members from *P. myriotylum* are found in the MCL_06066 group, which is unique to *P. myriotylum* (Table S2). GH62 members encode $\alpha$-l-arabinofuranosidases that can remove arabinose decorations on xylan (19). The *P. myriotylum* genome encoded the largest number of pectinases, with most of these annotated as GH28 endopolygalacturonase or PL3_2 pectate lyases (Table 2). The 16 putative CE5 family cutinase (CUT)/acetyl xylan esterase (AXE) enzymes from *P. myriotylum* are part of a single ortholog group along with seven cutinases/AXEs from *P. arrhenomanes* and nine cutinases/AXEs from *P. aphanidermatum* (MCL_00899).

Necrosis and peptide 1 (Nep1)-like protein (NLP), crinkler (CRN), and elicitin-like (but not RxLR) virulence factors were more abundant in *P. myriotylum* than in other *Pythium* species. There appears to be a similar size expansion of NLPs in *P. oligandrum*, but this is likely a separate event from that of the 19 NLPs from *P. myriotylum*, as the *P. oligandrum* NLPs appear to be type II NLPs (20). Strikingly, CRN type effectors were approximately 10-fold greater in number in *P. myriotylum* than in every other *Pythium* plant pathogen, with 61 CRNs annotated in the *P. myriotylum* genome (Table 3). In total, 100 elicitin-like proteins were annotated in *P. myriotylum*, but the increase over the number in other *Pythium* spp. was smaller than that found for the CRN effectors. There are seven genes annotated as RxLR type effectors in *P. myriotylum*, and the RxLRs do not appear more abundant in *P. myriotylum* (Table 3). Of note, *P. myriotylum* appears to have orthologs (Pm_g2845.t1 and Pm_g3051.t1) of the *Phytophthora infestans* SFI4 RxLR type effector (21).

**Tandem gene arrays contribute to the greater number of genes found in *P. myriotylum*.** The *P. myriotylum* genome was annotated with more genes (19,878) than all of the other *Pythium* genomes, with the exception of *P. guiyangense*, which is considered a hybrid species (Table 1). As our objective was to understand the pathogenicity of *P. myriotylum*, we examined virulence factor-related genes of interest, and it was noticeable from the gene identifier numbers that many of the virulence factor-related genes appeared to be located in tandem arrays. Tandem arrays were defined as two or more adjacent genes whose protein sequences have a level of similarity above a threshold (an E value of less than 1e−10 and a highest-scoring pair length greater than half the length of the shortest sequence). For example, the *P. myriotylum* genome contains 16 cutinase/AXE genes, and all of the cutinase/AXE genes appeared to be located in two almost adjacent tandem arrays: tandem array 1642 (*Pm_g15821* to *Pm_g15833*) and tandem array 1399 (*Pm_g15835* to *Pm_g15838*) (Table S5 and Fig. 4D). The numbers of genes found in tandem arrays in *P. myriotylum* and the other *Pythium* species were analyzed in more detail to understand whether this could have

**TABLE 2** Putative PCWDE content of *P. myriotylum* and other *Pythium* species[a]

| CAZy family (or subfamily) | Putative activities in CAZy family (or subfamily) | Putative substrate for enzyme | No. of PCWDEs in: | | | | | | | | | |
|---|---|---|---|---|---|---|---|---|---|---|---|---|
| | | | *P. myriotylum* | *P. aphanidermatum* | *P. arrhenomanes* | *P. guiyangense* | *P. insidiosum* | *P. irregulare* | *P. iwayamai* | *P. oligandrum* | *P. splendens* | *P. ultimum* |
| GH5_1 | EGL ($\beta$-1,4-endoglucanase) | Cellulose | 12 | 2 | 6 | 23 | 1 | 6 | 2 | 1 | 16 | 5 |
| GH5_12 | EGL ($\beta$-1,4-endoglucanase) | Cellulose | 1 | 1 | 1 | 4 | 1 | 2 | 0 | 1 | 5 | 5 |
| GH5_14 | $\beta$-glucosidase or $\beta$-1,3-glucanase | Cellulose | 29 | 4 | 9 | 10 | 8 | 5 | 6 | 19 | 8 | 9 |
| GH5_20 | EGL ($\beta$-1,4-endoglucanase) | Cellulose | 12 | 10 | 8 | 19 | 8 | 9 | 10 | 12 | 12 | 14 |
| GH6 | CBH (cellobiohydrolase) | Cellulose | 15 | 2 | 10 | 11 | 5 | 7 | 4 | 10 | 5 | 5 |
| GH7 | CBH (cellobiohydrolase) | Cellulose | 13 | 2 | 2 | 4 | 1 | 2 | 1 | 1 | 9 | 1 |
| GH131 | ML-EGL ($\beta$-1,3/$\beta$-1,4-endoglucanase) | Cellulose | 6 | 2 | 1 | 3 | 1 | 2 | 1 | 1 | 3 | 2 |
| AA16 | Monooxygenase | Cellulose | 4 | 4 | 2 | 5 | 2 | 4 | 1 | 1 | 5 | 4 |
| | | Cellulose total | 92 | 27 | 39 | 79 | 27 | 37 | 25 | 46 | 63 | 45 |
| GH10 | XLN ($\beta$-1,4-endoxylanase) | Xylan | 8 | 0 | 1 | 0 | 0 | 0 | 0 | 0 | 0 | 0 |
| GH11 | XLN ($\beta$-1,4-endoxylanase) | Xylan | 3 | 1 | 1 | 0 | 0 | 0 | 0 | 0 | 0 | 0 |
| GH62 | AXH ($\alpha$-l-arabinofuranosidases) | Xylan | 4 | 0 | 0 | 0 | 0 | 0 | 0 | 0 | 0 | 0 |
| CE2 | AXE (acetyl xylan esterase) | Xylan | 1 | 1 | 1 | 4 | 1 | 1 | 1 | 1 | 2 | 1 |
| | | Xylan total | 16 | 2 | 3 | 4 | 1 | 1 | 1 | 1 | 2 | 1 |
| GH5_9 | EXG (exo-1,3-galactanase) | Pectin | 0 | 0 | 5 | 1 | 2 | 0 | 0 | 0 | 0 | 0 |
| GH28 | PGA (endopolygalacturonase) | Pectin | 26 | 5 | 3 | 4 | 0 | 1 | 1 | 0 | 5 | 4 |
| GH35 | LAC ($\beta$-1,4-galactosidase) | Pectin | 1 | 1 | 1 | 2 | 2 | 1 | 1 | 1 | 1 | 1 |
| GH43_6 | Endo-$\alpha$-1,5-l-arabinanase | Pectin | 1 | 1 | 1 | 0 | 0 | 1 | 0 | 1 | 2 | 2 |
| GH53 | GAL ($\beta$-1,4-endogalactanase) | Pectin | 5 | 0 | 1 | 0 | 0 | 0 | 0 | 0 | 2 | 0 |
| GH78 | RHA ($\alpha$-rhamnosidase) | Pectin | 2 | 1 | 0 | 0 | 0 | 0 | 1 | 0 | 0 | 0 |
| CE13 | PAE (pectin acetylesterase) | Pectin | 9 | 3 | 3 | 6 | 4 | 3 | 3 | 7 | 6 | 4 |
| PL1_4 | Pectate lyase | Pectin | 6 | 7 | 1 | 8 | 1 | 5 | 2 | 4 | 10 | 12 |
| PL3_2 | Pectate lyase | Pectin | 25 | 14 | 3 | 12 | 4 | 6 | 2 | 12 | 32 | 15 |
| PL4 | Rhamnogalacturonan endolyase | Pectin | 1 | 0 | 2 | 2 | 0 | 2 | 2 | 0 | 6 | 2 |
| | | Pectin total | 76 | 32 | 20 | 35 | 13 | 19 | 12 | 25 | 63 | 42 |
| GH13_32 | AMY ($\alpha$-amylase) | Starch | 4 | 1 | 1 | 4 | 1 | 2 | 2 | 5 | 1 | 2 |
| GH15 | GLA (glucoamylase) | Starch | 4 | 2 | 1 | 5 | 4 | 1 | 1 | 2 | 3 | 2 |
| | | Starch total | 8 | 3 | 2 | 9 | 5 | 3 | 3 | 7 | 4 | 4 |
| CE5 | CUT (cutinase)/AXE (acetylxylan esterase) | Cutin, xylan | 16 | 7 | 6 | 0 | 0 | 0 | 0 | 0 | 0 | 0 |
| GH32 | Multiple activities within family | Inulin | 1 | 1 | 1 | 5 | 0 | 1 | 1 | 0 | 1 | 0 |
| GH1 | Multiple activities within family | Multiple substrates | 22 | 3 | 8 | 9 | 1 | 7 | 6 | 7 | 15 | 8 |
| CE1 | Multiple activities within family | Multiple substrates | 10 | 4 | 3 | 4 | 3 | 3 | 1 | 12 | 1 | 2 |
| GH2 | Multiple activities within family | Multiple substrates | 2 | 1 | 1 | 2 | 0 | 1 | 1 | 1 | 1 | 1 |
| GH5 | Multiple activities within family | multiple substrates | 3 | 0 | 1 | 0 | 0 | 0 | 0 | 0 | 0 | 0 |
| GH3 | Multiple activities within family | Multiple substrates | 10 | 7 | 7 | 19 | 4 | 8 | 6 | 8 | 9 | 7 |
| GH12 | Multiple activities within family | Multiple substrates | 3 | 0 | 2 | 0 | 0 | 0 | 0 | 0 | 0 | 0 |
| GH31 | Multiple activities within family | Multiple substrates | 6 | 6 | 4 | 9 | 3 | 5 | 4 | 7 | 3 | 5 |
| GH30_1 | Multiple activities within family | Multiple substrates | 9 | 0 | 4 | 11 | 4 | 6 | 5 | 9 | 11 | 7 |
| | | Overall total | 274 | 93 | 101 | 186 | 61 | 91 | 65 | 123 | 173 | 122 |

[a]The plant cell wall-degrading enzymes (PCWDEs) were annotated using dbCAN2 (65). Table S6 summarizes all of the CAZy families.

**TABLE 3** Summary of effector or other virulence factor-related gene families in *P. myriotylum* and other *Pythium* species[a]

| Category | No. of genes: | | | | | | | | | |
|---|---|---|---|---|---|---|---|---|---|---|
| | *P. myriotylum* | *P. aphanidermatum* | *P. arrhenomanes* | *P. guiyangense* | *P. insidiosum* | *P. irregulare* | *P. iwayamai* | *P. oligandrum* | *P. splendens* | *P. ultimum* |
| Aspartyl proteases | 56 | 30 | 32 | 86 | 32 | 22 | 23 | 18 | 66 | 40 |
| Cathepsin propeptide inhibitors | 5 | 3 | 3 | 9 | 9 | 5 | 5 | 4 | 4 | 5 |
| Chitinases | 0 | 4 | 3 | 6 | 10 | 3 | 2 | 16 | 2 | 3 |
| Crinkler (CRN) effectors | 61 | 5 | 6 | 11 | 0 | 1 | 2 | 25 | 4 | 4 |
| Cysteine-rich secretory proteins | 27 | 11 | 9 | 28 | 12 | 10 | 7 | 15 | 20 | 19 |
| Elicitins and elicitin-like proteins | 100 | 41 | 43 | 152 | 67 | 48 | 36 | 29 | 59 | 44 |
| IgA peptidases | 11 | 2 | 4 | 6 | 4 | 3 | 5 | 2 | 6 | 1 |
| Kazal-type protease inhibitors | 23 | 14 | 12 | 48 | 34 | 15 | 10 | 13 | 19 | 16 |
| Necrosis-inducing proteins (NLPs) | 19 | 4 | 5 | 2 | 0 | 4 | 4 | 17 | 9 | 7 |
| Papain family cysteine proteases | 41 | 19 | 23 | 78 | 37 | 23 | 26 | 18 | 35 | 25 |
| RxLR effectors | 7 | 3 | 5 | 2 | 2 | 4 | 5 | 14 | 11 | 4 |
| Subtilase proteases | 92 | 30 | 33 | 70 | 11 | 32 | 40 | 45 | 48 | 43 |
| Trypsin and trypsin-like proteins | 160 | 40 | 44 | 118 | 43 | 48 | 37 | 62 | 43 | 36 |
| Xylanase inhibitors | 0 | 1 | 1 | 3 | 0 | 1 | 1 | 0 | 1 | 0 |
| Total for species | 602 | 207 | 223 | 619 | 261 | 219 | 203 | 278 | 327 | 247 |

[a]The putative plant cell wall-degrading enzymes (PCWDEs) are shown in Table 2.

**A**

| Species | Total genes | Tandem clusters | Genes in tandem clusters | % Total genes | Average number of genes per cluster |
|---|---|---|---|---|---|
| *P. myriotylum* | 19,878 | 1,666 | 5,099 | 25.7 | 3.06 |
| *P. oligandrum* | 15,007 | 1,164 | 3,631 | 24.2 | 3.12 |
| *P. splendens* | 17,350 | 1,204 | 3,245 | 18.7 | 2.70 |
| *P. ultimum* | 15,290 | 848 | 2,214 | 14.5 | 2.61 |
| *P. guiyangense* | 30,943 | 1,660 | 3,950 | 12.8 | 2.38 |
| *P. iwayamai* | 14,869 | 742 | 1,593 | 10.7 | 2.15 |
| *P. insidiosum* | 14,962 | 684 | 1,516 | 10.1 | 2.22 |
| *P. aphanidermatum* | 12,312 | 526 | 1,185 | 9.6 | 2.25 |
| *P. irregulare* | 13,805 | 452 | 1,026 | 7.4 | 2.27 |
| *P. arrhenomanes* | 13,805 | 394 | 858 | 6.2 | 2.18 |

**B**

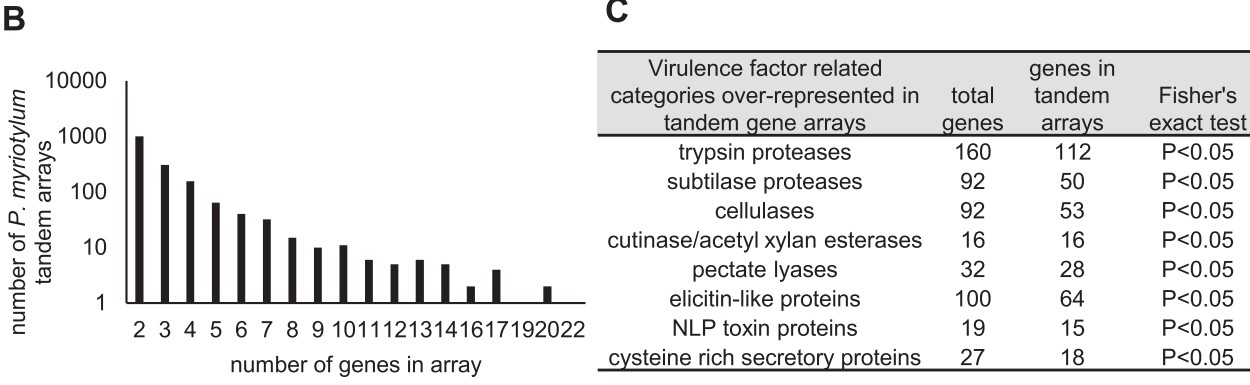

**C**

| Virulence factor related categories over-represented in tandem gene arrays | total genes | genes in tandem arrays | Fisher's exact test |
|---|---|---|---|
| trypsin proteases | 160 | 112 | P<0.05 |
| subtilase proteases | 92 | 50 | P<0.05 |
| cellulases | 92 | 53 | P<0.05 |
| cutinase/acetyl xylan esterases | 16 | 16 | P<0.05 |
| pectate lyases | 32 | 28 | P<0.05 |
| elicitin-like proteins | 100 | 64 | P<0.05 |
| NLP toxin proteins | 19 | 15 | P<0.05 |
| cysteine rich secretory proteins | 27 | 18 | P<0.05 |

**D**

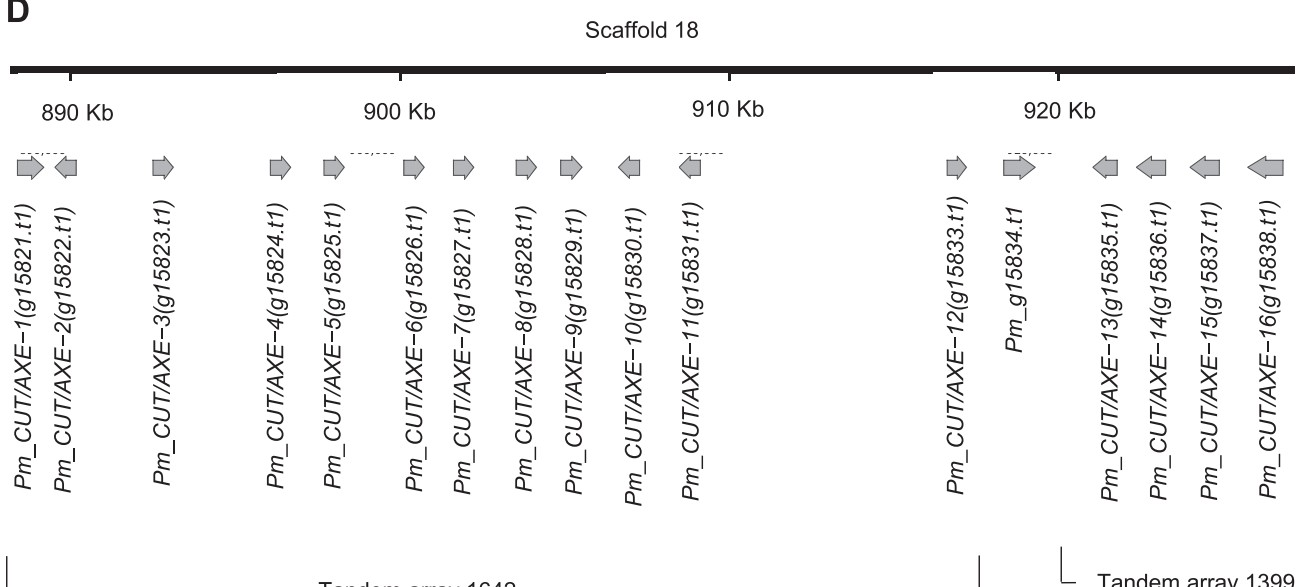

**FIG 4** Tandem gene arrays contribute to the greater number of genes found in *P. myriotylum*. (A) Comparison of tandem arrays in *P. myriotylum* with other *Pythium* species; (B) numbers of tandem arrays in *P. myriotylum* and numbers of genes found in these tandem arrays; (C) virulence factor-related categories overrepresented in tandem arrays in *P. myriotylum*; (D) an example of tandem arrays in *P. myriotylum* genome showing the arrays containing the *CUT/AXE* genes.

contributed to the greater number of genes in *P. myriotylum* than in the other *Pythium* species. *P. myriotylum* had the highest number of tandem gene arrays (1,666) and the highest number of genes found within tandem arrays (5,099) in comparison to other *Pythium* species (Fig. 4A). This trend whereby *P. myriotylum* has the

highest number of tandem arrays and genes within tandems is not because *P. myriotylum* has more genes than most of the other species, because when normalized for total gene number, the percentage of genes in *P. myriotylum* found within tandem arrays (27%) was also higher than that of all of the other *Pythium* species. The highest percentage among all of the other *Pythium* species was 24% in *P. oligandrum*, and the highest percentage in a plant pathogen was 19% in *P. splendens* (Fig. 4A). The majority of the tandem arrays contained two genes, and 22 genes was the highest number found within a single tandem array (genes in this array were annotated with the Pfam domain PF01328_Peroxidase, family 2) (Fig. 4B). Similar to the cutinases/AXEs, several other virulence factor categories were overrepresented in tandem arrays, including other polysaccharide acting activities (cellulases and pectate lyases), proteases (subtilases and trypsin proteases), elicitin-like proteins, NLP toxin proteins, and cysteine-rich secretory proteins (Fig. 4C).

There was substantial divergence among members of the tandem arrays, where only 95/1,666 of the tandem arrays have 100% average identity. The average protein identity between members of tandem arrays ranges from 23% to 100%, and the cumulative average of these percentage identities is 62% (Table S5). In an analysis of the selective pressure using the average ratio of nonsynonymous to synonymous evolutionary changes (*dN/dS*) of members of a tandem array, most of the tandem arrays (1,655) were under purifying selection (*dN/dS* ratio $< 1$), and only 11 were under diversifying selection (*dN/dS* ratio $> 1$).

To support the occurrence of the tandem arrays by gene duplication events, we used NOTUNG analysis of the type I NLP toxin proteins. Of the 19 NLPs in *P. myriotylum*, 15 of these are located in tandem arrays (tandem arrays 44, 1435, and 1614) (Table S5). NOTUNG uses phylogenetic analysis that compares gene and species trees and attempts to explain discordance between the gene and species trees. The discordance between the gene and species trees can be used to infer whether differences in gene content between species are due to gene gain (e.g., by duplication or horizontal gene transfer) or gene loss events. Here, the analysis supported the occurrence of many gene duplication events in the tandem arrays of the NLPs (Fig. S2). For example, in the cluster of the tree containing the well-known NLP from *P. aphanidermatum*, NLP$_{PYA}$ (PYAP24304), there are six *P. myriotylum* NLPs from a single tandem array (Pm_g13401 to Pm_g13406), and the gene gain in *P. myriotylum* here compared to that of the other two *Pythium* species is indicated by the NOTUNG analysis to have occurred via putative gene duplication events (Fig. S2). In the NOTUNG analysis, the gene loss events were far fewer than the duplication events (data not shown) in *P. myriotylum*, indicating that the duplication events are unlikely to be ancestral.

**Cross-isolate conservation of infection-upregulated pattern of virulence factor genes.** The ginger-infecting *P. myriotylum* SWQ7 and SL2 isolates (12) were analyzed by transcriptomics during infection of detached ginger leaves. We used ginger leaves as our model infection system to identify which genes in *P. myriotylum* could contribute to the successful infection of ginger. *P. myriotylum* is a soilborne pathogen, and generally, plant-pathogenic *Pythium* species are considered to enter plant tissue by penetration and to grow between and through plant cells (22). Mycelial mats of the *P. myriotylum* SWQ7 and SL2 isolates were used to form a sandwich with detached ginger leaves, and after 24 h, the leaves containing the ingressed mycelium were sampled for transcriptomics analysis for comparison with the mycelial mats before infection of the ginger leaves. From both of the isolates in a principal-component analysis (PCA) of the transcriptomic data, there was a clear separation between the transcriptome replicates from the mycelial control condition and those from infection of the ginger leaf (Fig. 5A). In SWQ7 and SL2, there were 2,110 and 2,513 genes, respectively, upregulated during infection of ginger. Approximately half of the genes upregulated in each isolate were also upregulated in the other isolate (Fig. 5B). The number of downregulated genes and the proportion shared between the two isolates were similar to those of the upregulated genes. These up- and downregulated genes likely include those directly responding to the presence of the ginger leaf and those upregulated due to

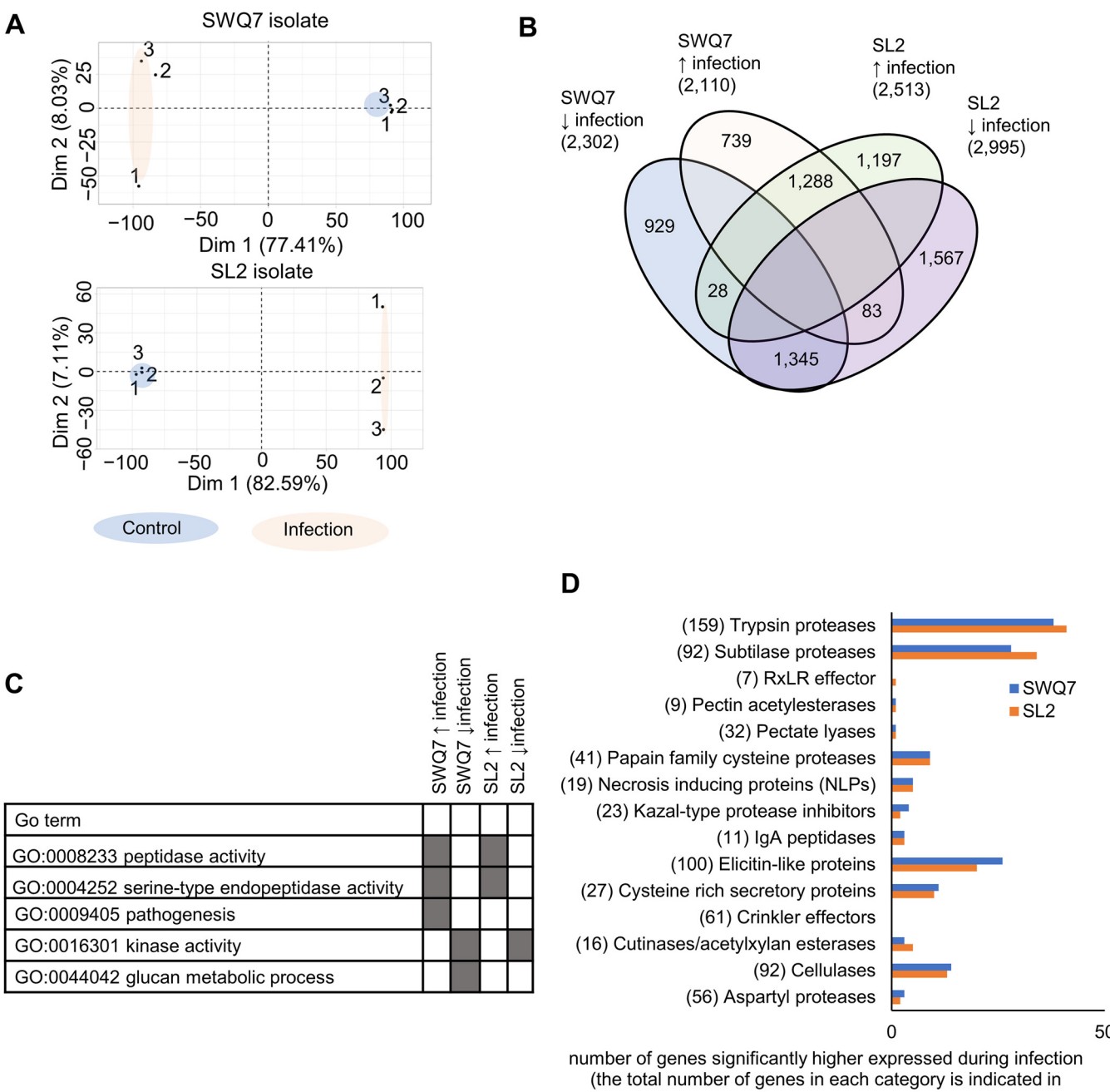

**FIG 5** Overview of transcriptional changes in *P. myriotylum* isolates SWQ7 and SL2 during infection of ginger leaves. (A) Principal-component analysis of replicate samples; (B) Venn diagram of the numbers of higher and lower expressed genes in SWQ7 and SL2 and during infection of ginger leaves; (C) selected terms from the GO enrichment analysis of the differentially expressed genes; (D) total number of genes induced in particular virulence factor-related categories.

differences in the growth environment of the ginger leaf compared to the growth on the V8 juice medium control.

There was an indication of a major role for proteases in the infection by both isolates whereby the GO term for peptidase activity (GO:0008233) was found to be enriched in both the SWQ7 and SL2 genes upregulated during infection of ginger (Fig. 5C). GO:0009405, the GO term for pathogenesis (of another organism), was enriched in the SWQ7 genes upregulated during infection of ginger, and this was due to upregulated putative elicitin-like proteins, the only ones encoded by genes in the *P. myriotylum* genome annotated with this GO term for pathogenesis. The largest number of virulence

factor-related genes upregulated were annotated as proteases, with ~100 protease-encoding genes upregulated.

Between 13 and 14 putative cellulase-encoding genes out of a total of 92 were up-regulated during infection of ginger leaves, and these cellulases may be involved in degrading the cellulose in the plant or *P. myriotylum* cell wall or both. Despite the large number of 11 xylanases in *P. myriotylum*, only a single xylanase was significantly upregulated in SWQ7 during infection of ginger leaves and none were in SL2. Surprisingly few of the pectinase-encoding genes were induced during infection of ginger leaves: for example, no GH28 endopolygalacturonases were induced or even expressed above a low level of expression (>10 fragments per kilobase per million [FPKM]), and a single pectin lyase (PL) Pm_g20951.t1 was induced from the PL sub-family PL3_2 in both isolates. Upregulation of putative cutinases/AXEs was more prominent, with three and five upregulated in the *P. myriotylum* SWQ7 and SL2 isolates, respectively (Fig. 5D). A subset of the NLP toxin proteins was highly induced during infection of ginger, with the same three NLPs induced to >100 FPKM in both of the *P. myriotylum* isolates (Fig. 5D). Of the 100 elicitin-like proteins that were annotated in *P. myriotylum*, approximately a quarter of these were induced during infection of ginger, with another quarter showing lower expression during infection compared to the control condition in both of the isolates (Fig. S3). Although the number of putative CRN effectors at 61 in the *P. myriotylum* genome was hugely expanded compared to that in other *Pythium* species genomes, none of the CRN effectors were significantly induced during infection of ginger. Only one of the putative RxLR effectors was induced in the *P. myriotylum* SL2 isolate during infection of ginger leaves, with a moderate level of expression (Table S7).

The comparative genomic analysis and transcriptomics during infection identified many interesting expansions of virulence factor-related gene families, and the activities of those related to plant cell wall degradation and host cytotoxicity were investigated.

***P. myriotylum* plant cell wall-degrading enzyme expansion did not correlate with growth.** We hypothesized that there would be better growth of *P. myriotylum* on plant biomass substrates for which the *P. myriotylum* genome encoded more enzymes for their degradation. Growth profiling on different carbon sources is a useful tool to relate genes required for the degradation of the polymeric carbon sources to the gene content of the respective genomes. Growth on monosaccharides alongside the polysaccharides can indicate whether an inability to catabolize the monomers that compose the polymers could contribute to the limited growth on a polymer apart from the lack of expression of the required polysaccharide-degrading enzymes. In general, there was no clear support for our hypothesis of a positive correlation between growth on various polysaccharides, and the higher number of genes encoding PCWDEs to degrade these polysaccharides in *P. myriotylum* than in the other *Pythium* species analyzed (Fig. S4). This was determined by the strength of growth on the carbon sources and taking into account any growth on the control of agarose medium without the addition of a carbon source. For example, there was not any better growth of either of the *P. myriotylum* isolates on two xylan substrates than of the *P. aphanidermatum*, *P. ultimum*, and *P. oligandrum* isolates (Fig. S4). These three other species contained only one (*P. aphanidermatum*) or no (*P. oligandrum* and *P. ultimum*) endoxylanases (Table 2). Growth on D-xylose (the monomeric unit of xylan), indicating the ability to catabolize D-xylose, was comparably weak for all four species. The growth was more similar in appearance to that of the control without an added carbon source than to that of the D-glucose cultures (Fig. S4A). The most noticeable difference between the four species was better growth of the *P. ultimum* isolates on a subset of sugars than of isolates of the other three species at the time point of 5 days. For the *P. myriotylum*, *P. aphanidermatum*, and *P. ultimum* isolates, the trends in growth when scored after 3 days were comparable across the different carbon sources, with the strongest and most consistent growth for both isolates of these species on the complex substrates of V8 juice, guar gum, and the powdered ginger rhizomes. The growth of the *P. oligandrum* isolates are more difficult to compare with that of the isolates of the other three *Pythium*

spp. since *P. oligandrum* grows with fewer aerial hyphae. In contrast, at the later time point of 5 days, when the images in Fig. S4A were taken, growth of *P. ultimum* isolates was as strong on another complex substrate starch and on the sugars D-glucose, D-fructose, and sucrose.

**Highly upregulated *P. myriotylum* putative cutinase encodes a functional esterase.** An interesting PCWDE activity that was encoded by more genes in *P. myriotylum* was cutinase/AXE (CAZy family CE5) (Table 2). As mentioned earlier, the 16 putative cutinase/AXE genes from *P. myriotylum* are located adjacent to each other in tandem gene arrays. Three of these cutinase/AXE genes (*Pm_CUT/AXE-4* [*Pm_g15824.t1*], *Pm_CUT/AXE-7* [*Pm_g15827.t1*], and *Pm_CUT/AXE-8* [*Pm_g15828.t1*]) were highly upregulated in *P. myriotylum* during infection of ginger leaves (Fig. 6A). Cutinases from oomycetes had not been heterologously produced previously for characterization of their enzymatic activities; therefore, to investigate the enzymatic activities of these putative cutinase/AXEs, Pm_CUT/AXE-8 was heterologously produced in *Pichia pastoris,* and the activity was measured toward *para*-nitrophenol ester-linked substrates that are substitutes for cutin monomer or shorter-chain fatty acids. For example, *p*NP-palmitate (*p*NPP) is a substitute for ester-linked 16-hydroxypalmitic acid, one of the major monomer units found in $C_{16}$-cutin (23). The culture supernatant from the Pm_CUT/AXE-8-expressing strain displayed approximately 20-fold-higher esterase activity than the background level from the empty-vector control (Fig. 6C). There was a significant preference from the purified Pm_CUT/AXE-8 for hydrolysis of longer-chain-length substrates, and the higher activity toward *p*NPP was found at lower temperatures and at neutral to mildly alkaline pH conditions (Fig. 6D). The three-dimensional (3D) structure of Pm_CUT/AXE-8 was modeled using two cutinase protein structures from *Aspergillus oryzae* CutL1 (24) and *Trichoderma reesei* Tr cutinase (25), and it has a catalytic triad formed from Ser, Asp, and His residues; additionally, Thr and Gln are also present in the active site of Pm_CUT/AXE-8, and there are two disulfide bridges present in the structure (Fig. S5).

**Multiple *P. myriotylum* NLPs have necrosis-inducing activity.** Virulence factors are critical determinants of whether a pathogen can successfully infect a host, and given that *P. myriotylum* is a necrotrophic pathogen, one of the foci for analysis of activities was those virulence factors or microbe- or pathogen-associated molecular patterns (MAMPs/PAMPs) with likely potential to induce necrosis in the host. We investigated whether *P. myriotylum* NLPs could cause necrosis in *N. benthamiana*, which as well as being a model plant system, tobacco is also another host besides ginger for *P. myriotylum*. NLPs are found in bacteria, fungi, and oomycetes, can be toxic to plant cells, and were comprehensively reviewed recently by Seidl and Van den Ackerveken (26). The necrosis-inducing ability of NLPs is generally limited to dicots (26). Previously, it was shown that the sphingolipid molecule to which NLPs bind, leading to toxicity, was commonly found in dicot plants such as *N. benthamiana*, used here for agroinfiltration experiments, but was generally different in monocots, which prevented the cytotoxicity (27). The primary classification system for NLPs is based on the number of cysteine residues with type I-containing two-cysteine residues and type II-containing four-cysteine residues. NLPs can be either toxic or nontoxic and can also function as a PAMP in some plant species, as well as potentially having functions unrelated to plant pathogenicity since NLPs are also found in, e.g., plant saprotrophs.

There are 19 putative NLP toxin proteins (all type I) in *P. myriotylum*, and all except one of these appear to contain key residues and motifs previously shown to be important for necrosis-inducing activity (Fig. 7B). Given the necrotrophic lifestyle of *P. myriotylum*, along with the presence of the residues and motifs for necrotic activity, our hypothesis was that at least some of the *P. myriotylum* NLPs would cause necrosis in *N. benthamiana* leaves. Agroinfiltration of *N. benthamiana* leaves with strains expressing a subset of *P. myriotylum* NLPs was used to test the cytotoxicity of the NLPs. As a positive control, an *Agrobacterium* strain expressing the *Phytophthora infestans* elicitin INF1, which causes necrotic lesions in *N. benthamiana* leaves similar to those caused by NLPs, was used, and as a negative control, an *Agrobacterium* strain expressing green fluorescent protein (GFP) was used. The proteins encoded by two of these highly induced NLP genes (*Pm_NLP-1* and *Pm_NLP-4*) have

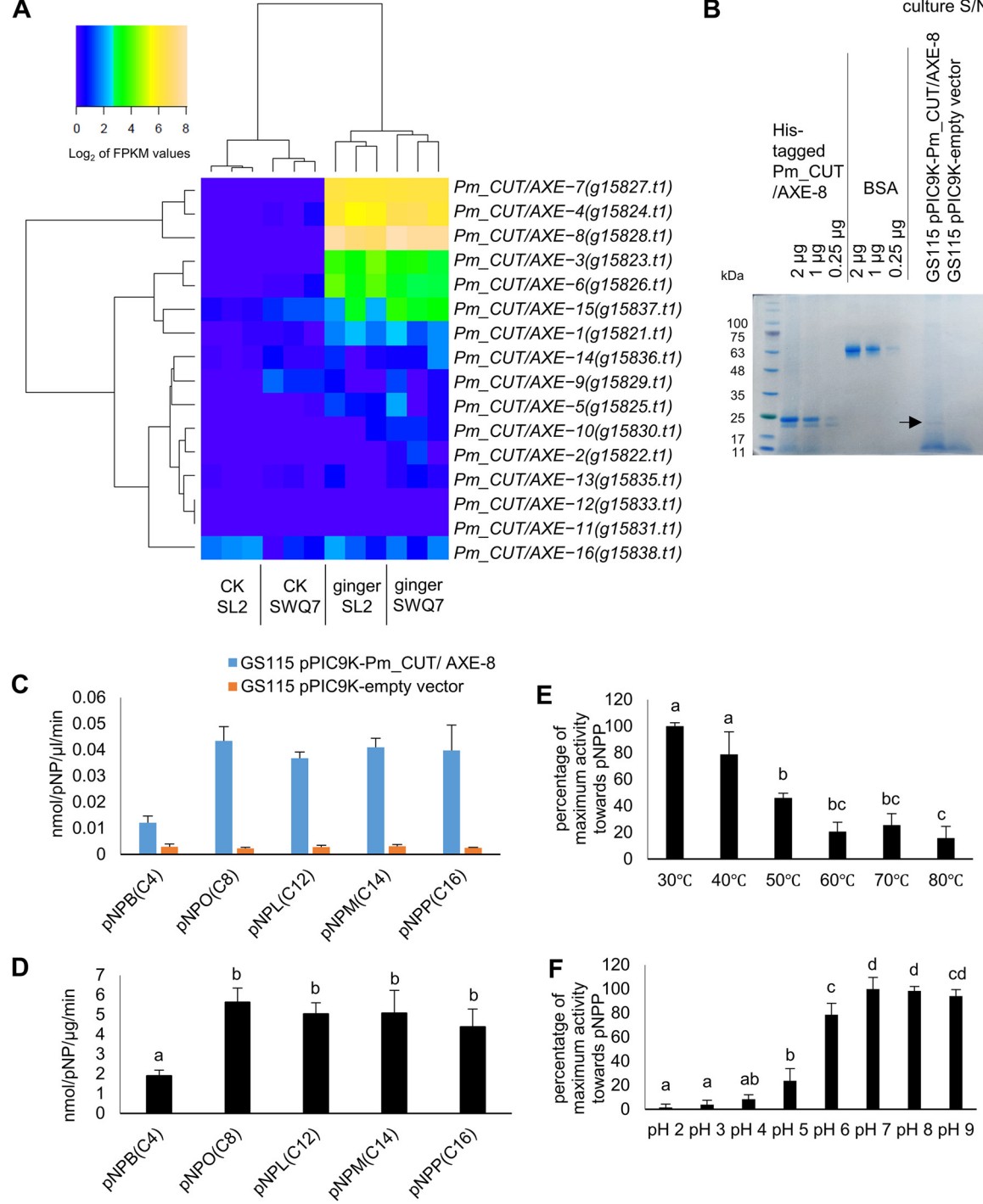

**FIG 6** Expression of *P. myriotylum* putative cutinases during infection of ginger leaves and activity of Pm_CUT/AXE-8. (A) Hierarchical clustering heat map showing the $\log_2$ FPKM expression levels of *P. myriotylum* putative cutinases before and after infection of ginger leaves; (B) PAGE gel showing the purified protein and culture supernatants before purification from the *P. pastoris* GS115 strain expressing Pm_CUT/AXE-8 (arrow) and the empty vector strain (the predicted size of the protein is 21 kDa); (C and D) esterase enzyme activity toward *para*-nitrophenol (*p*NP) substrates of various chain lengths from the culture supernatant (C) and from the purified Pm_CUT/AXE-8 protein (D); (E and F) temperature (E) and pH (F) optima for esterase enzyme activity from the purified Pm_CUT/AXE-8 protein toward *p*NPP. The error bars represent standard deviation ($n = 2$ for assays with culture supernatants and $n = 3$ for assays with purified protein). *p*NPB = *para*-nitrophenyl butyrate; *p*NPO, *para*-nitrophenyl octanoate; *p*NPL, *para*-nitrophenyl laurate; *p*NPM, *para*-nitrophenyl myristate; *p*NPP, *para*-nitrophenyl palmitate. The activity levels were compared using analysis of variance (ANOVA) with a *post hoc* Tukey's test; bars that contain the same letter are not significantly different from each other ($P < 0.05$).

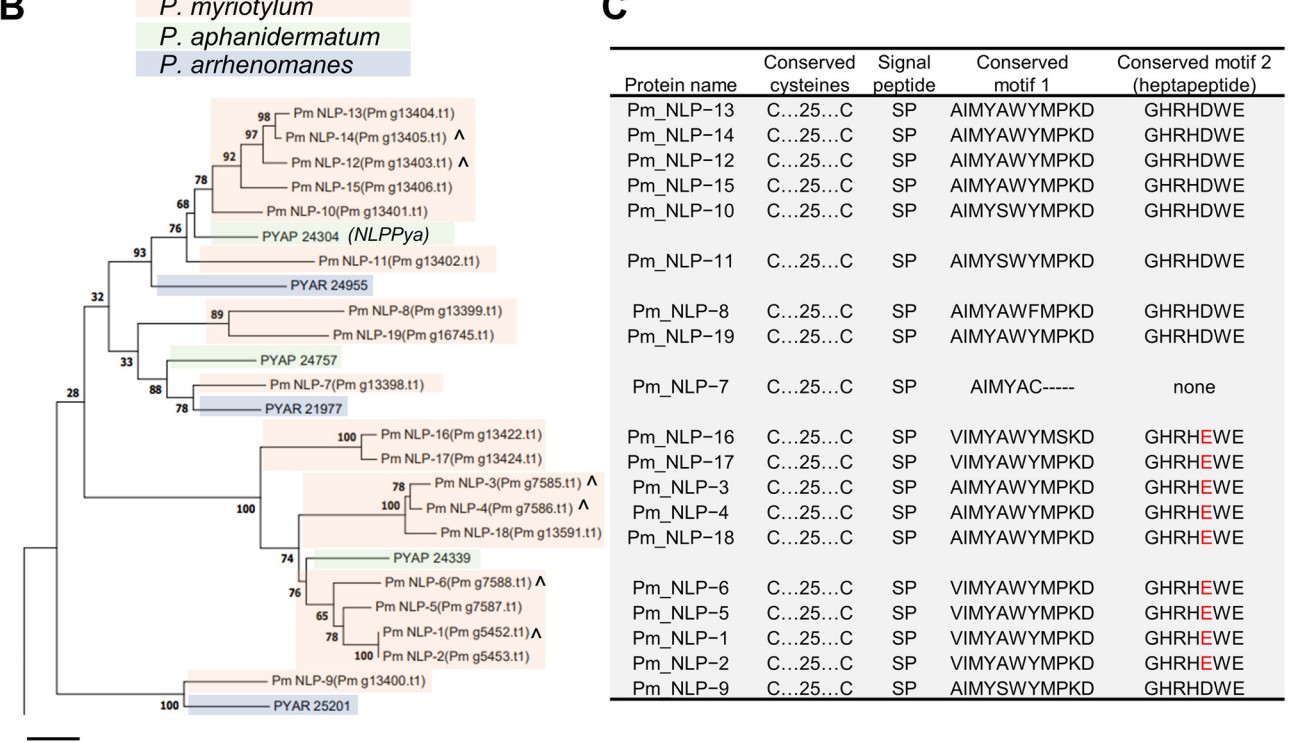

**FIG 7** Expression during infection of ginger of *P. myriotylum* NLP toxin proteins and necrosis-inducing activity. (A) Hierarchical clustering heat map showing the log₂ FPKM expression levels of *P. myriotylum* NLPs before and after infection of ginger leaves along with whether selected effectors induce

necrosis-inducing activity in *N. benthamiana* leaves, where the timing of the necrosis-inducing activity was similar to that of the positive control, INF1 (28). The proteins encoded by four other NLP genes that were not significantly induced during infection of ginger (*Pm_NLP-3*, *Pm_NLP-6*, *Pm_NLP-12*, and *Pm_NLP-14*) also had necrosis-inducing activity in *N. benthamiana* leaves. Pm_NLP-3 and Pm_NLP-14 had necrosis-inducing activity that appeared similar to that of the INF1 control, whereas Pm_NLP-12 and Pm_NLP-14 both had necrosis-inducing activity that appeared to be weaker than that of the INF1 control as it occurred later (Fig. 7A). Pm_NLP-12 and Pm_NLP-14 clustered with NLP$_{Pya}$ in the phylogenetic analysis, and Pm_NLP-10 to Pm_NLP-15, which occur in a tandem duplication, appear to be orthologs of the characterized NLP$_{Pya}$ from *P. aphanidermatum* (29). For the *P. myriotylum* NLPs analyzed here by agroinfiltration, their production in *N. benthamiana* leaves was confirmed by Western blotting, where the hemagglutinin (HA) tag on these proteins was detected (Fig. S6).

## DISCUSSION

Here, we have shown how *P. myriotylum* has a larger arsenal of virulence factor-related genes than other *Pythium* plant pathogens and how a subset of these are up-regulated during infection of ginger and can have potential functions in pathogenicity.

The reason why *P. myriotylum* has the most virulence factor-related genes may be related to its broad host range with particular subsets of the genes required for pathogenicity in different hosts. *P. myriotylum* has the largest number and proportion of predicted secreted proteins. A previous study suggested a relationship between secretome size (estimated from genome annotation) and fungal lifestyle; e.g., fungi with a biphasic lifestyle can have proportionally larger secretomes (30). As far as we are aware, there is no evidence for a lifestyle of *P. myriotylum* other than as a necrotrophic plant pathogen. Expansion of the number of genes in virulence factor categories may be related to a mechanism to increase gene expression levels other than by activating transcription factors.

During infection of ginger, there was upregulation of a subset of virulence factor-related genes. The upregulation of only a subset may be related to timing or host preference whereby the noninduced genes are required at a different time or for a different host. Evasion or activation of the immune response by a virulence factor may also be a contributing factor in the subset of genes that is induced (as reviewed recently by Chen et al. [31]). The physicochemical properties of the host environment may be another factor in the induction of a subset of genes. *P. myriotylum* has both pectin lyases and galacturonan hydrolyases (Table 2), and there was only a single pectate lyase induced. In the fungus *Aspergillus niger*, it has been suggested that its genome mainly encodes pectin hydrolyases, not lyases, because *A. niger* acidifies the medium (32). The heterologously produced Pm_CUT/AXE-8 had a preference for neutral and mildly alkaline pH levels (Fig. 6), and this pH-dependent activity profile could be beneficial for pathogenesis, as the apoplastic space is reported to alkalinize during plant-pathogen interactions (33). Of note, the cutin polymer is found in the epidermis of aerial tissues but not in root tissue, through which infection by soilborne pathogens such as *P. myriotylum* can occur. Instead, degradation in older roots of the cutin-like suberin polymer which cutinases can also degrade may contribute to *P. myriotylum* infection of root tissues (34).

The lack of a positive correlation between the growth on xylan and the presence of xylanase-encoding genes in the genome of *P. myriotylum* compared to the other *Pythium* spp. (see Fig. S4 in the supplemental material) could be explained by several factors. There may be a lack of gene induction of the xylanase-encoding genes during growth on the xylan polysaccharides. During infection of the ginger leaves, which contain xylan, only a single endoxylanase gene (*Pm_g13123.t1*) was induced and only to a

**FIG 7** Legend (Continued)

necrosis in *N. benthamiana* leaves (the number of replicate leaves is indicated in parentheses); (B) maximum likelihood phylogenetic tree of *P. myriotylum* NLPs (scale bar, 0.1 amino acid substitution per site) and NLPs from two relatively closely related *Pythium* species, i.e., *P. aphanidermatum* and *P. arrhenomanes*; (C) information on the presence of a signal peptide and key residues and motifs known from previous studies to be important for necrosis-inducing activity.

relatively low expression level. A lack of appropriate catabolic pathways for xylose catabolism may be another explanation, as the weak growth of *P. myriotylum* on xylose compared to other carbon sources such as glucose suggests a lesser ability to catabolize xylose. Previously, poor growth on D-xylose was seen for the *P. ultimum* strain DAOM BR144 (35). The *P. myriotylum* endoxylanases may be used primarily to degrade xylan polymers to gain access to other polymers which contain sugars that it can metabolize (e.g., the glucose in cellulose) and to damage host tissues as part of the necrotrophic attack. This is somewhat surprising, given that xylan is generally the most abundant hemicellulosic polymer in angiosperm plants and would provide *P. myriotylum* with a plentiful energy source. Catabolism of xylose via the pentose catabolic pathway is common in saprotrophic fungi (36). Also, it is noteworthy that most of these isolates from the other species are not those whose genomes have been sequenced, and the assumption is made that the isolates used in our study have PCWDE repertoires that match the genomes of the sequenced strains.

The role of the induced NLP toxin proteins in ginger infection remains to be elucidated, as NLPs are generally not known to induce necrosis in monocots such as ginger. The necrosis-inducing ability of NLPs is generally limited to dicots (26). Previously, it was shown that the sphingolipid molecule to which NLPs bind, leading to toxicity, was commonly found in dicot plants such as *N. benthamiana*, which was used for the agro-infiltrations (Fig. 7), but that this molecule was generally different in monocots and prevented the cytotoxicity (27). The induction of the NLPs during infection of ginger may be part of a more generic plant infection-related gene upregulation (i.e., not specific to ginger), as *P. myriotylum* is a broad-host-range pathogen, including of dicots such as tobacco (*Nicotiana tabacum*) (11), which is closely related to *N. benthamiana*, in which the tested NLPs induced necrosis (Fig. 7). In *Phytophthora capsici*, the 11 NLPs selected for functional analysis led to various levels of toxicity on *N. benthamiana* leaves, with three of the NLPs leading to strong toxicity and the other eight leading to weaker toxicity, observed as chlorotic rather than necrotic areas on the leaves (37). Of the six NLPs tested from *P. myriotylum*, they all could induce necrosis, and this is perhaps in contrast to the NLPs analyzed in some other species, where NLPs that lacked necrosis-inducing activity were also found, e.g., in *Phytophthora sojae* (38). Although the total number of NLPs annotated in the *P. myriotylum* genome is relatively large, this seems to be the result of a relatively recent genome expansion by tandem duplication, in comparison with other *Pythium* spp. (Fig. 7B), and this may limit the evolution of different functions (necrosis-inducing versus non-necrosis-inducing NLPs). The large number of NLPs may also be due to the fact that *P. myriotylum* is a pathogen of dicot species, where the possession of these toxic NLPs is an advantage to the necrotrophic lifestyle. Also, the annotation of NLPs in *P. myriotylum* is based on the Pfam domain for NLP and may be more conservative than overall homology-based approaches that may also identify pseudogenes or those lacking part of the functional domains. Furthermore, in an analysis of the selective pressure on the *P. myriotylum* NLPs, the ratio of nonsynonymous to synonymous substitutions (dN/dS) suggests that all *P. myriotylum* NLPs appear to be under purifying selection (Table S9), and this would correlate well with the fact that all six of the tested *P. myriotylum* NLPs could induce necrosis in *N. benthamiana* leaves. The identification of *P. myriotylum* NLPs with necrosis-inducing activity suggests that these NLPs could be promising targets, as it was recently shown that NLPs from other plant pathogens are targets for inhibition by novel phytopharmaceutical agents (39). An alternative explanation for the maintenance of an expanded set of NLPs may be related to NLP functions other than as a virulence factor, whereby the presence of NLPs in non-plant-pathogenic plant saprotrophs suggests an alternative role in ecological niche establishment.

The genome and transcriptome results correlate well with previous findings on *P. myriotylum* compared to other *Pythium* species. The importance of pectin degradation for infection by *P. myriotylum* was shown by the loss of pectin from cocoyam (*Xanthosoma sagittifolium*) occurring early during infection by *P. myriotylum* (40). Correlating well with

the importance of pectin, the genome of *P. myriotylum* encodes a large number of pectin-degrading enzymes. Also supporting a prominent role for PCWDEs in infection by *P. myriotylum* was where in bitter ginger (*Zingiber zerumbet*), a relative of *Zingiber officinale*, transcription factors involved in cell wall fortification were upregulated in response to *P. myriotylum* infection (17). PCWDEs are abundant in the genome of *P. myriotylum* and induced during infection of the ginger leaves (Table 2 and Fig. 5). The importance of proteases to *P. myriotylum* infection seems to be supported by the presence of a protease-inhibiting antimicrobial peptide from the rhizome of bitter ginger (41). Proteases are a prominent feature of the genome and infection-induced transcriptome of *P. myriotylum* (Fig. 3 and 5).

There was evidence from the genome of substantial retrotransposition machinery whereby 2,779 gene models with significant matches to transposon sequences were identified. It was previously noted by Lévesque et al. (42) and in references cited therein that the presence of DNA methyltransferases can inhibit repeat expansion by methylation of promoter elements (43). More recently and supporting this, when an $N^6$-adenine methyltransferase domain-containing (DAMT) protein was knocked out in *P. sojae*, transposable elements were more active (44). In *P. myriotylum*, the numbers of DNA methyltransferases are also low based on Pfam domain annotation, with only one protein containing the $N^6$-adenine methyltransferase (PF10237) domain, one containing an MT-A70 SAM-binding subunit (PF05063) domain, and none from other Pfam DNA methyltransferase domain categories (PF00145, PF02086, and PF05869). The *P. myriotylum* genome repeat content of 31% is relatively high compared to what was reported for other *Pythium* species; e.g., for *P. guiyangense*, a repeat content of 6% was reported (18). Taken together, the substantial level of retrotransposition machinery in the *P. myriotylum* genome, the low number of DNA methyltransferases, and the high repeat content indicate that repeat expansion is an important feature of the genome dynamics of *P. myriotylum*.

Technical reasons can potentially be a factor when comparing genome sequences from different technology platforms. There are potential biases due to different sequencing technologies that can lead to artefactual differences between the more recently sequenced third-generation *Pythium* genomes (*P. guiyangense* and *P. oligandrum*) and the genomes sequenced with older technologies (Sanger and second-generation sequencing). For example, in an updated assembly of the *Phytophthora ramorum* genome incorporating long sequencing reads, the repeat content of the genome almost doubled (45). Third-generation sequencing platforms can more easily assemble repetitive parts of genomes than earlier technologies, but this is unlikely to explain all the differences between *P. myriotylum* in repetitive content, as other *Pythium* genomes also assembled from third-generation sequencing technology have lower repeat content than *P. myriotylum*. Differences in when gene calls were made and differences in repeat annotation and subsequent repeat masking of genomes can be contributory factors, but these are unlikely to affect the level of expansion of the virulence factor-related gene categories reported for *P. myriotylum*. We did not reannotate the genomes of the other *Pythium* species using our pipeline for *P. myriotylum* because this can complicate comparisons with the previous literature on these other *Pythium* species (e.g., if a new set of gene identifiers is generated).

In conclusion, *P. myriotylum* has the most extensive arsenal of virulence-related genes among the broad-host-range necrotrophic *Pythium* plant pathogens analyzed, and tandem duplications contributed to this expansion. The expansion of these virulence-related genes likely contributes to the success of *P. myriotylum* as a broad-host-range pathogen, although factors other than expansion of gene families also contribute, as *Pythium* spp. with fewer virulence-related genes also have broad host ranges, e.g., *P. aphanidermatum*. The large number of virulence factor genes and the heterozygosity underline the importance of genome resequencing of isolates infecting particular hosts (our high-quality genome assembly can form the reference for these resequencing efforts).

## MATERIALS AND METHODS

***Pythium* species and isolates.** *P. myriotylum* SWQ7 (CGMCC no. 21459) and SL2 (CGMCC no. 21956) were isolated from infected ginger rhizomes from locations 100 km apart in Shandong province, one of the major ginger-growing regions in China (12). The internal transcribed spacer (ITS) region sequence (GenBank accession no. MT482756.1) and the *CoxII* region sequence (MT505384.1) of SWQ7 are 99.86% (720/721) and 99.81% (518/519) similar, respectively, to the sequences from the *P. myriotylum* reference strain CBS254.70 (10). The ITS region of the rRNA and *CoxII* region sequences of the *P. myriotylum* SL2 isolate were identical to those of the SWQ7 isolate. The other *Pythium* spp. used in growth profiling were from various sources, and their identities were confirmed by sequencing at least the ITS region using primers ITS1 and ITS4, described previously (46). The *P. oligandrum* isolate GAQ1 and the *P. ultimum* isolate S001 were obtained from soil samples from China. The *P. ultimum* isolate G001 and *P. aphanidermatum* isolates HBT1 and HPW9 were obtained from infected ginger rhizomes from China. The other *P. oligandrum* isolate, CBS530.74, has been described previously (47).

**Growth cabinet pathogenicity test on ginger plants.** Ginger plants derived from tissue culture were used in pot trials to compare their susceptibilities to the *P. myriotylum* SWQ7 and SL2 isolates by following the method of Le et al. (48), with the following several modifications: the inoculum was prepared using wheat seeds, the medium used was 10% V8 juice agar, the plants were about 3 weeks old after deflasking, and the plants were grown in three replicate 100-mL pots in a growth cabinet set at 25°C with a 12-h light and dark cycle with a light intensity of 60%. Periodically, the ginger plants were given a water-soluble fertilizer (NPK 20-20-20+TE). Plants were monitored daily for disease development based on above-ground symptoms by applying the following scale from Stirling et al. (49): 0, plants remain green and healthy; 1, leaf sheath collar is discolored and lower leaves turned yellow; 2, plants alive, but shoots either totally yellow or dead; and 3, all shoots dead. The disease index (DI) was calculated separately for the plants on each of the three replicate sets on separate shelves in the growth chamber (each shelf contained 10 each of control plants, SWQ7-infected plants, and SL2-infected plants), using the formula $DI = \sum10_{i\,=\,1} X_i/30$, where DI is the disease index rating from 0 to 1, $X_i$ is the disease rating of the $i^{th}$ replicate (from 1 to 10), and 30 is the number of replicates when multiplying by the highest rating scale of 3. Koch's postulates were demonstrated by reisolating *P. myriotylum* from surface-sterilized, infected roots and confirming the species identity by sequencing the ITS region.

**DNA preparation, genome sequencing, and resequencing.** For genomic DNA extraction from SWQ7 and SL2, mycelium from liquid cultures inoculated with zoospore-purified isolates was used. DNA from the SWQ7 isolate was sequenced using PacBio continuous long-read sequencing (Sequel system instrument) and Illumina short-read sequencing (HiSeq X instrument), and the DNA from the SL2 isolate was sequenced using only Illumina short-read sequencing.

**Genome assembly of *P. myriotylum*.** Illumina sequencing reads were adapter and quality trimmed using Trim_galore! (v0.64). Genome size and heterozygosity levels were estimated by generating k-mer count histograms from Illumina reads using Meryl (https://github.com/marbl/meryl), which were used as input for GenomeScope 2 (50). PacBio reads were corrected, trimmed, and assembled using Canu (v2.0) (51). Contigs arising from haplotypic duplication were identified and removed using purge_dups (52). The assembly was polished using two rounds of Pilon (v1.23) (53) with Illumina reads. SSPACE (54) was used to scaffold the assembly further, and gaps were closed with GapFiller (55). Scaffolds originating from contamination were identified using CAT (56) and removed. Genome assembly statistics were calculated using QUAST (v5.0.2) (57). Genome completeness was estimated using BUSCO (v3) (58) with the "Alveolata-Stramenopiles" data set. K-mer spectral analysis and consensus quality (QV) estimation were performed using Merqury (59). *De novo* repeat family identification was performed using RepeatModeler (v2.0.1) (60) and RepeatMasker (v4.1.1), but note that the genome itself was not masked before gene prediction and instead gene models representing putative transposable elements were excluded at a later stage, as described below.

**Gene prediction and functional annotation.** Transcriptome sequencing (RNA-Sequencing) reads were aligned against the genome assembly using STAR (v2.7.3) (61), and the subsequent alignment was used to train BRAKER2 (v2.1.4) (62) with Augustus (63) for gene prediction. Gene models representing putative transposable elements were excluded if they shared significant sequence similarity (BLASTp E value cutoff of 1e−10) to the TransposonPSI library (http://transposonpsi.sourceforge.net/). Completeness of the gene set was estimated using BUSCO (v3) (58) with the "Alveolata-Stramenopiles" data set. Genes were functionally annotated using InterProScan (64). CAZymes were annotated using dbCAN2 (65). Secreted proteins and transmembrane proteins were predicted using SignalP (v3) (66) and TMHMM (v2.0) (67), respectively. For a protein to be considered secreted, it had to have a positive prediction from SignalP, a hidden Markov model (HMM) S probability value of ≥0.9, an NN Ymax score of ≥0.5, an NN D score of ≥0.5, and no transmembrane domains downstream of the predicted signal peptide cleavage site. Tandemly duplicated genes were identified using BLASTp (68). Tandem clusters were defined as two or more adjacent genes that hit each other in a BLASTp search with an E value of less than 1e−10 and a highest-scoring pair (HSP) length greater than half the length of the shortest sequence. Inference of gene duplication in the type I NLPs from *Pythium* spp. was performed using NOTUNG (69), where trees were constructed using IQ-TREE (70). Effector or other virulence factor-related categories were annotated primarily using Pfam or other InterProScan domains similar to those described previously (20, 71). For particular effector categories, these required further or different annotation approaches: the crinkler (CRN) proteins were annotations using an HMM and the presence of the L[FY]LA[RK] and HVLVXXP motifs, and the RxLR proteins were annotated using the modified regular expression model described by Ai et al. (72). To test the selective pressure acting on the *P. myriotylum* type I NLP sequences, the yn00 program (73) implemented in the PAML package was used to calculate the dN/dS (nonsynonymous/synonymous) ratios.

**Comparative *Pythium* genomics.** For the comparative *Pythium* genomics, the genomes of the following *Pythium* species were used: *P. aphanidermatum* (5), *P. arrhenomanes* (5), *P. guiyangense* (18), *P. insidiosum* (74), *P. irregulare* (5), *P. iwayamai* (5), *P. oligandrum* (75), *P. splendens* (76), and *P. ultimum* var. *ultimum* DAOM BR144 (42). The predicted protein sequences were downloaded from NCBI, from the Oomycete Gene Order Browser (http://www.ogob.ie/gob/data.html), or from other sources (e.g., obtained from coauthors). The functional annotations on the gene models for each of the other *Pythium* genomes were carried out at the same time as the annotations of the *P. myriotylum* gene models using InterProScan (64). Gene families were identified by performing all-versus-all BLASTp searches with an E value cutoff of 1e−10, followed by Markov clustering using MCL (77) with an inflation value of 1.5. Core genes are those that are present in all 10 genomes, dispensable genes are those that are present in 2 or more genomes, and unique genes are those present in only 1 of the 10 genomes. Gene ontology enrichment analyses were performed using Fisher's exact test with the Bonferroni correction for multiple testing using GOATOOLS (78). Corrected *P* values of <0.05 were considered significant. For the GO enrichment analysis for comparison of the genomes, the proteins from all 10 *Pythium* genomes (168,168 proteins) were assigned GO IDs with 64,459 proteins (approximately 50%) receiving at least one GO term. Of the 19,878 *P. myriotylum* proteins, 7,131 had at least one GO term associated. When Fisher's exact test was run, the total protein complement with GO annotations was considered the "population." The *P. myriotylum* genes were considered the "study." Fisher's exact test asks the question of whether there are particular GO terms in *P. myriotylum* that are enriched ("e") or purified ("p"; underrepresented) relative to the population (i.e., the other *Pythium* species being considered). Only terms that were significantly enriched or purified after the Bonferroni correction were considered. The analysis was performed for *P. myriotylum* versus all *Pythium* species, plant-pathogenic *Pythium* species, and non-plant-pathogenic *Pythium* species.

For the phylogenomic analysis, BUSCO analysis was performed using the eukaryotic data set and identified 145 BUSCO proteins that are present as single copies in at least 11 of the 12 species. Each BUSCO family was individually aligned using MUSCLE (79). Alignments were then trimmed using trimAl (80) with the "automated1" parameter and concatenated together, resulting in a supermatrix alignment of 68,028 amino acid residues. A maximum likelihood phylogeny was generated using IQ-TREE (70) under the "LG+F+R5" model, which was the best fit model according to ModelFinder (81), with 100 bootstrap replicates. For the phylogenetic tree of the NLPs, the amino acid sequences were aligned using MAFFT (https://www.ebi.ac.uk/Tools/msa/mafft/), and a maximum likelihood phylogenetic tree was generated in MEGA7 (82) using the JTT model (gamma distribution with invariant sites). All positions with less than 95% site coverage were eliminated, and 1,000 bootstrap replicates were performed.

**Experimental conditions for transcriptomics experiment.** Fresh cultures of the SWQ7 and SL2 isolates on 10% V8 agar were used to inoculate 10% V8 liquid medium. The outer two-thirds of the colony on a single plate was sectioned into ∼0.5-cm by ∼0.5-cm agar squares, and all of the agar squares were transferred to a 9-cm petri dish containing 10% V8 liquid medium. The petri dishes containing the agar squares inoculum were incubated in the dark at 25°C for 36 h without shaking. The mycelium was harvested (whereby the agar squares were separated from the mycelial mat), washed twice in water, and either collected and flash frozen in liquid nitrogen to use as the control noninfected condition or placed between two ginger leaves. Healthy ginger leaves ∼10 cm in length were taken from young plants grown in a greenhouse. Each infection replicate consisted of six ginger leaves forming three side-by-side sandwiches between the mycelium (washed mycelium harvested from one plate was used for each infection replicate). The leaf-and-mycelium sandwiches were kept moist by placing them in a plastic film-covered tray on wetted filter paper where the base of the leaf that was wrapped in wetted tissue was in contact with the wetted filter paper layer. The leaf-and-mycelium sandwiches were incubated at 25°C in the dark for 24 h. For harvesting from the sandwiches, the parts of the leaves not contacting the mycelium were cut off, and then the free mycelium between the leaves was gently peeled away (note that this mycelium was not used for RNA extraction). The leaf containing the closely attached mycelium was gently blotted to remove excess liquid, flash frozen in liquid nitrogen, and stored at −80°C before the RNA was extracted.

**RNA extraction, RNA-Seq library preparation, sequencing, and data analysis.** The infected-leaf samples and mycelium control samples were ground in liquid nitrogen, and the RNA was extracted using TRIzol reagent. Residual DNA was removed from the RNA samples by using a Dnase treatment. The RNA integrity was verified using a Bioanalyzer (Agilent), and the RNA purity was measured using a spectrophotometer. A 2 × 150 bp paired library was prepared using the TruSeq stranded mRNA LTSample prep kit (Illumina) and sequenced using the Illumina HiSeq X platform. Raw data (raw reads) were processed using Trimmomatic (83). The reads containing poly A/T tail sequence and low-quality reads were removed to obtain clean reads. Downstream analysis of the cleaned reads was performed using the following software tools at https://usegalaxy.eu (84). The cleaned reads were mapped using HISAT2 Galaxy version 2.1.0 (85) to the SWQ7 genome assembly described earlier. All reads with an alignment quality of ≥10 were counted using HTSeq-count Galaxy version 0.9.1, where the union mode was used to handle reads overlapping more than one feature and nonuniquely mapping reads were counted on all the features to which they mapped (86). Differential gene expression analysis of the count data was performed using DESeq2 Galaxy version 2.11.40.6, where the default Cook's distance cutoff was used for outlier filtering (87). The criteria for a differentially expressed (DE) gene were FPKM of >10 in one condition, DESeq $P_{adj}$ value of <0.05, DESeq $\log_2$ fold change [FC] of greater than 1 or less than −1. GO enrichment analysis of the DE genes was performed using BiNGO (88). The FPKM values were calculated by expressing the counts for a gene per kilobase of gene length and per million mapped counts to all genes. Hierarchical clustering was performed using the $\log_2$ FPKM values in the R statistical environment using the gplots package with the Euclidean distance and complete linkage

options selected. PCA was performed on the moderately and highly expressed genes (where the FPKM was >10 in at least one sample) using FactoMineR version 1.39 (89) in R.

**Growth profiling of *Pythium* spp. on a range of plant biomass-related carbon sources.** Carbon sources were prepared in an *Aspergillus* minimal medium (AMM) adjusted to pH 6 (90). The mono- and disaccharides were prepared as stock solutions at concentrations between 0.25 and 1 M, filter sterilized, and added to the autoclaved AMM-agarose at a final concentration of 25 mM. The complex polysaccharides were added to the AMM-agarose at a final concentration of 1% (wt/vol) before autoclaving at 121°C for 15 min. Ultrapure agarose (Invitrogen; catalog no. 16500-100) was used at a 1.5% (wt/vol) concentration for the growth profiling experiments. The following 17 carbon sources were used: D-glucose, D-fructose (Macklin; catalog no. D809612), D-galactose (Sigma; V900922), D-mannose (Macklin; M6146), D-xylose (Sangon; XBO998), L-arabinose (Macklin; L824031), L-rhamnose monohydrate (Macklin; L817271), cellobiose (Macklin; C6182), sucrose (Macklin; S818046), D-galacturonic acid hydrate (Macklin; D807796), xylan from corn cob (Macklin; X823251), xylan from beechwood (Megazyme; P-XYLNBE), guar gum (Macklin; G810488), soluble starch from potato (SCR; 10021318), pectin (Macklin; P816453), $\alpha$-cellulose (Macklin; C804601), and ginger rhizome extract powder (Laiwu Manhing Vegetables Fruits Corporation). Precultures of the *Pythium* isolates were prepared using filtered V8 juice agar. Triplicate 6-cm petri dishes for each carbon source (and a control without an added carbon source) were inoculated with 2-mm$^2$ plugs from the growing edge of the colony. The cultures were incubated at 25°C in the dark, and after 3 days, the thickness/strength of the mycelial growth was scored, and after 5 days, images of the cultures were taken.

**Heterologous production of putative cutinase and assays for enzyme activity.** A codon-optimized version of Pm_CUT/AXE-8 was inserted into the pPIC9K vector using the EcoRI and NotI sites, and the SalI linearized vector was used to transform the *P. pastoris* GS115 strain. The strain was transformed, and transformants were selected with the G418 antibiotic by using standard protocols. After initial confirmation of production of the protein from the culture supernatant, large-scale methanol-induced production was performed from inoculation of a seed culture in buffered glycerol complex medium (BMGY) into a 1-L culture with buffered methanol complex medium (BMMY) at 30°C at 250 to 300 rpm. Methanol at 1.5% (wt/vol) concentration was used, and the culture supernatant was harvested after 120 h. For purification, 1 L of the culture supernatant was precipitated with 48% saturated ammonium sulfate overnight to remove part of the pigment and protein and then centrifuged at 12,000 rpm for 10 min. The precipitated protein was resuspended with 20 mM Tris-HCl and 50 mM NaCl (pH 8.0) and dialyzed three times. The dialyzed protein was passed through a DEAE-Sepharose ion exchange column, the protein was eluted with 20 mM Tris-HCl–300 mM NaCl, and the purity and concentration of the protein were detected using PAGE gels with Coomassie blue staining. ImageJ analysis of the purity indicated that the protein was approximately 85% pure. The eluted protein was dialyzed once with 20 mM Tris-HCl–50 mM NaCl, pH 8.0. The purified enzyme was stored in aliquots at $-80°C$.

For assays of esterase activity using *para*-nitrophenol-linked substrates, the standard assay buffer contained 20 mM Tris-HCl (pH 7.5), 10 mM NaCl, 50 mM sodium taurodeoxycholate, 1 mM *p*NP substrate, 0.1% gum arabic, and purified protein or culture supernatant in a total volume of 100 $\mu$L in a 96-well plate. Reaction mixtures were incubated at 37°C and monitored continuously using a 96-well plate reader at 405 nm for hydrolysis of the *p*NP substrates. The amount of *para*-nitrophenol released was quantified using a standard curve on each plate. The enzyme activity was expressed as nmol of *para*-nitrophenol (*p*NP) released per min per $\mu$g of protein or per $\mu$L of culture supernatant. Abbreviations for the substrates used were as follows: *p*NPB, *para*-nitrophenyl butyrate; *p*NPO, *para*-nitrophenyl octanoate; *p*NPL, *para*-nitrophenyl laurate; *p*NPM, *para*-nitrophenyl myristate; and *p*NPP, *para*-nitrophenyl palmitate. The *p*NP substrates were prepared as 20 mM stocks in Triton X-100–isopropanol (2:3). For the assays at different temperatures, the standard assay conditions described above were used with a PCR machine at 30°C, 40°C, 50°C, 60°C, 70°C, and 80°C, where the reaction mixtures were heat inactivated at 95°C for 5 min before transfer of the mixtures to a 96-well plate to measure absorbance. For the assays at different pH levels, Britton-Robinson buffer (40 mM) was used to make buffered solutions from pH 2 to pH 9, and this buffer was supplemented with 10 mM NaCl–50 mM sodium taurodeoxycholate and 0.1% (wt/vol) gum arabic. The reactions at different pH levels were terminated with 100 $\mu$L of 0.25 M Na$_2$CO$_3$. The assays were repeated in duplicate (with the culture supernatants) or triplicate (with the purified protein) with two or three technical replicates in each assay.

**Protein modeling and structural analysis.** Pm_CUT/AXE-8 was modeled using the I-TASSER (Iterative Threading ASSEmbly Refinement) server. A total of five models were generated with threading. The best structural model was determined based on comparison of confidence (C) scores, template modeling (TM) scores, and cluster densities. Modeled protein structure was visualized with PyMol software and compared to the crystal structure of the cutinases from *Aspergillus oryzae* (CutL1) (24) and *Trichoderma reesei* Tr cutinase (25) as structural homologues. Active-site residues and disulfide-forming Cys residues were predicted based on sequence and structural alignments. Average scaled hydropathy scores were calculated and plotted with the Bioconductor Charge and Hydropathy Vignette (91). Evolutionary conserved residues were predicted through ConSurf (92) analysis of the Pm_CUT/AXE-8 structure.

**Agroinfiltrations of *N. benthamiana* with *P. myriotylum* NLP toxin proteins.** A subset of the *P. myriotylum* NLP toxin proteins (NLP-1, NLP-3, NLP-4, NLP-6, NLP-12, and NLP-14) were amplified from genomic DNA (gDNA) of the SWQ7 isolate and cloned into the vector pGR107-3×HA using the ClonExpress II one-step cloning kit (Vazyme). The sequence of the NLPs amplified included the sequence encoding the signal peptide and excluded the stop codon. The GFP sequence was amplified from the pTOR-GFP plasmid to generate a strain to use as a negative control for the agroinfiltrations. The primers used in this work are listed in Table S10 in the supplemental material. The cloning reactions were transformed first into *Escherichia coli*, and the plasmids recovered were sequenced (Sangon Biotech) to

confirm the presence of the target sequence in the plasmid. These plasmids were used to transform the *Agrobacterium tumefaciens* GV3101 strain, which contained the helper plasmid pJIC Sa_Rep. The plasmids were recovered from the *A. tumefaciens* strains and sequenced to confirm the presence of the target sequence in the plasmid.

The *Agrobacterium* cultures were inoculated from a single colony in LB supplemented with 50 $\mu$g/mL kanamycin and 50 $\mu$g/mL rifampin, grown overnight at 28°C with shaking, and then diluted in 20 mL of LB with the same antibiotic concentrations to grow overnight again to reach an optical density at 600 nm ($OD_{600}$) of ~0.9. The cultures were centrifuged at 3,000 rpm at room temperature for 10 min. The cell pellet was washed with the infiltration buffer (10 mM $MgCl_2$, 10 mM morpholineethanesulfonic acid [MES], 150 $\mu$M acetosyringone). The cells were centrifuged again and resuspended in infiltration buffer to give an $OD_{600}$ of 0.5 and then incubated at 28°C in the dark without shaking for 3 h. Zones of ~1 to 2 cm on leaves from 5- to 6-week-old *N. benthamiana* plants were infiltrated on the abaxial side of the leaf using a needle-less syringe with ~50 $\mu$L of the *Agrobacterium* cell suspension. Separate zones on each leaf were infiltrated with strains of the following: one of the candidate *P. myriotylum* NLPs, pGR107-INF1-3×HA (a *P. infestans* elicitin [28]) as a positive control, pGR107-GFP-3×HA as a negative control, and the infiltration buffer. The infiltrated *N. benthamiana* plants were kept at ~25°C for 1 week and monitored for the appearance of necrosis. The infiltrations were repeated on at least 20 leaves.

Western blotting was used to confirm the production of the *P. myriotylum* NLPs in the *N. benthamiana* leaves. Infiltrated leaf zones were collected at several time points after agroinfiltration, flash frozen in liquid nitrogen, and stored at −80°C. Crude protein was extracted from the leaf zones by using a plant protein extraction kit (Solarbio; BC3720) according to the manufacturer's instructions. Equivalent volumes of the protein extractions were separated on SurePAGE 8-to-16 % Bis-Tris polyacrylamide gels (GenScript) using a dithiothreitol (DTT)-containing protein loading buffer (Coolaber; catalog no. SL1170) whereby the samples in the buffer were heated at 98°C for 8 min. A protein marker (Solarbio; PR1910) was used to indicate the size of the proteins. The proteins were transferred to an eBlot L1 polyvinylidene difluoride (PVDF) membrane (GenScript; L00735C) using an eBlot L1 fast transfer machine (GenScript; L00686). The membrane was blocked using a solution containing 5% skimmed milk. Primary antibodies raised against the hemagglutinin (HA) tag (Abmart; M20003) were used at a 1:5,000 dilution. The secondary antibody, also used at a 1:5,000 dilution, was goat anti-mouse IgG antibody conjugated with horseradish peroxidase enzyme (Abmart; M21001). A chemiluminescence imaging system (Tanon; 5200 Multi) was used to detect the chemiluminescent signal generated using the high-sig ECL Western blotting substrate (Tanon; 180-5001). To demonstrate equivalent loading of protein on the gel and transfer of protein to the membrane, Ponceau S staining was used.

**Data availability.** The sequencing data for the SWQ7 and SL2 isolates have been deposited at DDBJ/ENA/GenBank under BioProject accession no. PRJNA692555 and PRJNA701145, respectively. The genome assembly for SWQ7 was deposited as part of BioProject accession no. PRJNA692555. The RNA-Seq data set was deposited in the GEO database under accession no. GSE174557. All other relevant data are within the manuscript and its supplemental material files.

## SUPPLEMENTAL MATERIAL

Supplemental material is available online only.
**SUPPLEMENTAL FILE 1**, PDF file, 2.1 MB.
**SUPPLEMENTAL FILE 2**, XLSX file, 2.1 MB.
**SUPPLEMENTAL FILE 3**, XLSX file, 0.1 MB.
**SUPPLEMENTAL FILE 4**, XLSX file, 2.9 MB.
**SUPPLEMENTAL FILE 5**, XLSX file, 0.1 MB.
**SUPPLEMENTAL FILE 6**, XLSX file, 0.02 MB.
**SUPPLEMENTAL FILE 7**, XLSX file, 8.9 MB.
**SUPPLEMENTAL FILE 8**, XLSX file, 0.04 MB.
**SUPPLEMENTAL FILE 9**, XLSX file, 0.1 MB.
**SUPPLEMENTAL FILE 10**, XLSX file, 0.01 MB.

## ACKNOWLEDGMENTS

We acknowledge Suomeng Dong for the pGR107-3×HA vector and Daolong Dou for the gift of the *P. oligandrum* CBS530.74 isolate.

This work was financially supported by the National Natural Science Foundation of China (grant no. 32050410305), the China Agriculture Research System (grant no. CARS-24-C-01), the Jiangsu Agricultural Science and Technology Innovation Fund [grant no. CX(18)2005], and the China Postdoctoral Science Foundation (grant no. 2020M671388).

P.D. and D.Z. wrote the manuscript, analyzed the data, and supervised the experiments. D.S., F.C., and G.P. performed bioinformatics analysis and contributed to discussion and revision of manuscript. Y.C. carried out the growth profiling experiments. T.X. carried out the NLP agroinfiltration work. Q.Z. carried out the transcriptomics experiments. J.Z., J.L., and S.C. assisted in various experiments. J.M. and D.A.F. assembled and annotated the genome

and performed comparative genomic analysis. S.D., I.S.D., T.M.M.S., and N.W. contributed to discussion and revision of manuscript. G.B.A., H.O.S., and I.K. performed protein modeling analysis. L.W. managed the overall project and obtained funding.

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
