## [Reviewer comments · Microbiology Spectrum]

Microbiology Spectrum

Genome of *Pythium myriotylum* uncovers an extensive arsenal of virulence-related genes amongst the broad-host range necrotrophic *Pythium* plant pathogens

Paul Daly, Dongmei Zhou, Danyu Shen, Yifan Chen, Taiqiang Xue, Siqiao Chen, Qimeng Zhang, Jinfeng Zhang, Jamie McGowan, Feng Cai, Guan Pang, Nan Wang, Taha Majid Mahmood Sheikh, Sheng Deng, Jingjing Li, Okan Soykam, Irem Kara, David Fitzpatrick, Irina Druzhinina, Günseli Akcapinar, and Lihui Wei

Corresponding Author(s): Paul Daly, Jiangsu Academy of Agricultural Sciences

Review Timeline:

Submission Date:	November 15, 2021
Editorial Decision:	January 3, 2022
Revision Received:	June 29, 2022
Accepted:	June 30, 2022

Editor: Lindsey Burbank

Reviewer(s): Disclosure of reviewer identity is with reference to reviewer comments included in decision letter(s). The following individuals involved in review of your submission have agreed to reveal their identity: Michael Matson (Reviewer #3)

Transaction Report:

DOI: <https://doi.org/10.1128/spectrum.02268-21>

January 3, 2022

Dr. Paul Daly
Jiangsu Academy of Agricultural Sciences
Institute of Plant Protection
No. 50 Zhongling Street
Nanjing 210014
China

Re: Spectrum02268-21 (Genome of *Pythium myriotylum* uncovers the most extensive arsenal of virulence-related genes amongst the broad-host range necrotrophic *Pythium* plant pathogens)

Dear Dr. Paul Daly:

Thank you for submitting your manuscript to Microbiology Spectrum. The reviewers commented on the value of these genomic resources to the research community and noted several positive aspects of your study. However, there are some concerns that still need to be addressed regarding support for the conclusion that this *Pythium* species has an expanded virulence gene repertoire due to tandem duplications. Please carefully address the comments of the reviewers, with particular attention to details of the assembly methodology which could help the reader interpret your findings better. Thank you for your patience with the review process at a busy time of the year.

Link Not Available

Sincerely,

Lindsey Burbank

Journals Department
Reviewer comments:

Reviewer #1 (Comments for the Author):

The manuscript by Daly and colleagues reports on the genome sequencing, annotation, and analysis of *Pythium myriotylum*, a broad host range necrotrophic pathogen. It also reports on an expanded gene repertoire compared with other, related pathogens, and links these expansions to the occurrence of tandem gene arrays. Furthermore, the manuscript discusses the functional analysis of some virulence-factor related gene families. The objectives of the study are clearly stated, and the generated genome and transcriptome data will be of interest to the research community. However, I have several concerns related to the data and the drawn conclusion as well as the clarity of the manuscript.

A key observation is that the number of predicted protein-coding genes is large (23k) compared with other *Pythium* species. This number is indeed very high compared with gene numbers reported for most oomycetes thus far, and thus necessitates scrutiny. It seems that at least a considerable proportion of these genes are (remnants of) transposable elements, as the author reports on the enrichment of annotations associated with TEs in their results (L200ff). It is unclear if the gene annotation was based on the repeat-masked genome assembly or on the complete assembly. At any case, the authors need to remove genes that are likely related to transposons from their predicted gene set. Only then the notion of expanded gene sets compared with other *Pythium* species could be supported, especially if it would turn out that this difference is strongly driven by differences in repeat content. Furthermore, as transposons tend to cluster in oomycetes (see e.g., Haas et. al 2001), some of the tandem 'gene' clusters might in fact be due to repeat-islands rather than genuine tandem duplications of protein-coding genes. This therefore is a major technical limitation that needs to be addressed, as it affects multiple key messages of this manuscript.

One of the main deliverables to the research community is the genome assembly of *P. myriotylum* strain SWQ7. However, the manuscript reports very little details on the amount and quality of the data generated and of the assembly itself. For instance, the manuscript should include an indication of the yield, read N50, as well as (supplementary) figures on the k-mer distributions etc. to assess genome sizes. Given the complex nature of oomycete genomes, a k-mer spectrum as reported by KAT would be insightful, as it also reports on k-mers missing from the final assembly as well as k-mers present in multiple copies. As the genome is much bigger, details and an in-depth discussion on the number of duplicated BUSCOs (and later in the manuscript on duplicated genes in general) is essential. Lastly, as the assembly is reported as haploid, the authors need to better explain their strategy to move from a partial diploid assembly to a haploid representation, including how many contigs were removed/purged etc. All these essential information is lacking to judge the quality of the genome assembly, which is the basis of all the reported analysis.

The manuscript is not easy to follow and read and much essential information is reported in the M&M (e.g., on the genome assembly) or in the figure legends. More importantly, the motivation of specific experiments (e.g., the authors' hypothesis) and the explanation of the experiments, its results, and the authors conclusion are lacking throughout the manuscript. This, unfortunately, makes the overall manuscript very hard to comprehend. For instance, it remains unclear why the authors focused on the esterase and NLPs for their functional assays and what their expectations have been. If they hypothesized that the tested esterase could contribute to virulence and/or pathogenicity as hinted at in the results and discussion, then the performed experiments are not sufficient to address this notion. Knock-down and/or -out experiments and pathogenicity assays would need to demonstrate a contribution to virulence, which are very challenging in oomycetes. Similarly, it is unclear why the manuscript reports on growth profiles on different carbon sources, and it is anyway questionable if the limited carbon sources tested would be suitable to uncover correlations in growth and the presence and/or expansion of CAZymes. Lastly, one of the key limitations of the study, the impact of the genome assembly technology and gene annotation on the reported results is only addressed very late in the discussion. While I agree with the authors that the expansion of many gene families will likely not be impacted by erroneous gene annotation, the impact of the inclusion of transposable elements on the overall gene number and on the occurrence of tandem gene arrays needs to be addressed.

Additional concerns, comments, and suggestions:

L47: 'plant-virulent' - it is obvious what the manuscript attempts to say, but it sounds strange to refer to a strain isolated from diseased ginger rhizomes as being plant-virulent. I would suggest omitting the phrase.

L59: Throughout the manuscript it remains unclear how the observation of gene expansions in *P. myriotylum* will lead to knowledge-driven disease management strategies. The authors need to clarify this in the discussion or reconsider the conclusion of their work.

L66: 'Pythiums' 'Pythium species'

L84: 'Here we ...' I would suggest moving this sentence towards the end of the introduction and connected with the rest of the summary of this work.

L129: The period is too complex and difficult to follow.

L138: The statement is very relevant for the manuscripts narrative and thus not be placed in brackets.

L142: More details should be reported at this stage on the sequence technology and genome assembly strategy.

L147: What is meant by 'The final...'? The sentence seems grammatically not correct and necessitates further explanation. For instance, what do the authors mean by 'haplotypic duplications' and why the assembly from long-read was considered to be a diploid assembly.

L151: This sentence is grammatically not correct

L154: The authors need to provide more detail on their analysis rather than merely referring to it as 'phylogenetic analysis'. Why

was this analysis performed? On which marker was this tree based? Some of this information is obviously in the M&M but the manuscript becomes incomprehensible without the essential information.

L197: Could the statement 'which are often secreted' please be quantified by providing the number of secreted proteins?

L224. This sentence is grammatically not correct

L254: 'above a threshold' this is very vague and should be clarified

L262: 'While this trend...' I am unfortunately not able to follow the authors' argument. Could they please rephrase this sentence to clarify their argument?

L278: The definition of tree reconciliation is not entirely correct. NOTUNG, as many other tree reconciliation algorithms, aims to explain discordance between species and gene trees and not, as mentioned by the authors, whether gene content differences are due to gains. Based on the expansion of NLPs, and many other gene families, in *P. myriotylum*, it is to be expected that these are due to duplication. However, the interesting question not addressed by the authors is if these duplications are *P. myriotylum* specific or are ancestral and gene copies have been lost in sister species, which can be derived from the NOTUNG analyses. Furthermore, it would be relevant (see also comment below) if recent duplicates are organized in tandem arrays or not.

L292: It is unclear from the text what analysis has been performed.

L317, L319: 'Between 10 to 16...' 'Despite the large number...' how many cellulases or xylanases are encoded in the genome? To aid interpretation, the manuscript should report these numbers here too.

L332: 'relatively low levels' this is very vague. What is relatively low? What is high? The authors need to offer context.

L346: Remove the () around the explanatory statement as it is relevant to understand the experiment and its results.

L371: Three of the AXE genes were upregulated and, based on the gene code, seem to be localized in a single locus. What is the nucleotide similarity between these genes and could overall similarity between genes in these tandem arrays impact the gene expression analysis, e.g., due to multi-mapped RNAseq reads?

L393: The manuscript needs to properly introduce NLPs as well as the differences between type I/II, toxic/non-toxic as well as the expected phenotypes based on the positive control INF1.

L427: This statement necessitates an appropriate reference.

L438: 'The *P. myriotylum* infection strategy...' This statement is highly speculative, and it is not clear to me how this conclusion is supported by the presented data.

L446: 'A lack of ...' this statement could be further supported by checking the genome sequence for the presence of the corresponding enzymes.

L464: Another potential explanation for the occurrence of an expanded set of NLPs is the potential role of NLPs in niche establishment as NLPs have been commonly found to also occur in saprophytes.

L575: Which PacBio long-read sequencing platform was used for this study?

L586: How many corrections to the genome assembly have been performed by Pilon for each iteration? How many changes or uncertainties were retained after the 2nd iteration?

L692: The authors applied a cutoff of FPKM > 10 to differentiate between expressed and non-expressed genes. How has this cutoff been chosen, and how many genes are not expressed under this cutoff? Furthermore, on line 326 the manuscript mentions that three NLPs are highly expressed based on an FPKM of 100. If genes < 10 are considered non-expressed, then a FPKM of 100 does not seem to be particularly high. How does this relate to the expression levels of other genes in the genome?

Table 1: It is unclear how the species in Table 1 are ordered. If there is no specific rationale in the order, I would suggest considering to order based on the phylogeny shown in Figure 2.

Table 2: It is unclear why the sum of all proteins per category would be informative as it depends on the number of species considered. For clarity, the table should omit this column.

Figure 1: Panel B needs to report the individual datapoints underlying the barplots to enable readers to assess the spread of the

data. Furthermore, it would be useful if they figure could also show representative figures for different disease index to judge how DI relates to symptoms.

Figure 2: The authors could consider integrating some data from table 1 (e.g., genome size, repeat content, and/or gene content) into this figure.

Figure 3: To ease interpretation of the figure, panel A should report the ratio or the fold-change rather than the individual numbers.

Figure 4: To further support the frequent occurrence of tandem arrays in *P. myriotylum*, the authors should consider showing few example regions. Panel c should report the ratios, fold-changes, and p-values of the overrepresented categories.

Figure 5: It is counter intuitive that the dimensionality reduction of the transcriptomes of each isolate is reversed in the same panel. Furthermore, the vertical line in the panel for SL2 is not explained and unclear. Panel D should relate the number of up-regulated genes to the total number of genes in each category and each strain.

Figure 6: The color key for the heatmap is not sufficiently labeled. One of the replicates of the control for SL2 behaves different than the other replicates. To which extent does this impact the differential expression analysis (as mentioned in L300)? To obtain robust gene expression, the authors should consider repeating the expression experiment. The labels on top of the inoculation areas need to be moved to the side to aid visual inspection of the necrotic areas. Furthermore, the necrotic areas are hard to see and judge. The authors should complement the visual representation of the analysis with quantification of necrotic areas, for instance by using red light imaging (Villanueva et al. 2021). As mentioned for Fig. 4 the authors should consider showing a genomic locus of recently expanded NLPs to demonstrate how these evolved in tandem arrays, see e.g., Dong et al. 2012; Fig. 6.

Figure S2. A substantial number of proteins have both signal peptides as well as TMHMM. This suggests that either the protein is not a secreted ELL or that the SignalP and/or TMHMM prediction is incorrect. Furthermore, some ELL do not have a signal peptide. What does this suggest in terms of their biological function and/or gene annotation?

Reviewer #2 (Comments for the Author):

Daly and coworkers have isolated the highly virulent SWQY strain of *Pythium myriotylum* from spice ginger. In a GO enrichment analysis, hydrolase activity and DNA transposition was enriched in *P. myriotylum* compared to other *Pythium* species. Compared to other *Pythium* plant pathogen genomes, *P. myriotylum* has two to three times higher number of plant cell wall degrading enzymes. Many of the virulence genes were found in tandem arrays. NOTUNG analysis supported the presence of tandem arrays by gene duplication. Transcriptome of *P. myriotylum* infecting ginger, showed that proteases and plant cell wall degrading enzymes were upregulated. Genes identified in the comparative and transcriptomic analysis were shown to be related to pathogenicity (necrosis, and polysaccharide degradation) in infection and enzyme activity assays.

Correct statistical analyses were performed and shown in the figures or methods. However, these statistical analyses can also be mentioned in the main text

Results:

Line 132, please, state that a student's t-test was done and p-value for the disease index assay.

Line 152, mention that 145 single copy proteins were used for the phylogenomic analysis, and type of approach (Max Likelihood)

Line 184, indicate that Bonferroni correction applied to GO enrichment analysis p-values

Fig 5A, need higher resolution to see PCA plot

Line 364, indicate the statistical test (ANOVA)

Line 677-685, For differential expression analyses, it is better to use normalization methods of DESeq2 or EdgeR, median of ratios or TMM, rather than FPKM. The authors use DESeq2 which takes raw counts and normalizes the data so I am curious as to how they were able to work with FPKM.

Reviewer #3 (Comments for the Author):

This work does a great job summarizing the extensive repeat-driven diversification of the *Pythium myriotylum* genome and how it contributes to its success as a broad-range necrotrophic plant pathogen. Of particular note is the enrichment of tandem repeats

which appears to be driving the substantial increase in total gene count.

It is impressive to assemble such a repeat-laden oomycete genome sequence to a scaffold N50 of 1.6 Mb and only 185 total contigs. However, some statistics such as contig number, N50, and repeat percentage should be more explicitly stated in the text (around line 161), and repeat content is only ever mentioned in the entire manuscript in the discussion on line 512. This would be a useful statistic to include in table 1 in comparison to the other sequenced *Pythium* species. Additionally, the particular PacBio technology (HiFi reads??) needs to be stated in the methods (line 575).

In relation to the analysis of tandemly duplicated genes, one statistic that would be useful is the average % identity shared between members of each array. Are they often identical in sequence, or have they substantially diverged? A related question would be: what is the inferred timeline of these duplications based on sequence divergence? Are these recent, or are they broadly similar to divergence rates of other oomycetes? Are only the NLPs diverging rapidly compared to the rest of the genes (line 483)?

In relation to the analysis of the PCWDEs, my specific expertise is not in the profiling of gene activity and biochemistry. However, I believe the authors have done a thorough job exploring the interesting features of the extensive repertoire of *P. myriotylum* virulence factors, especially when experimentally confirming the activity of NLP proteins.

Staff Comments:

Preparing Revision Guidelines

Please return the manuscript within 60 days; if you cannot complete the modification within this time period, please contact me. If you do not wish to modify the manuscript and prefer to submit it to another journal, please notify me of your decision immediately so that the manuscript may be formally withdrawn from consideration by Microbiology Spectrum.

Authors have indicated that sequencing data of both isolates were deposited in GenBank

Comments to Authors:

Summary:

Daly and coworkers have isolated the highly virulent SWQY strain of *Pythium myriotylum* from spice ginger. In a GO enrichment analysis, hydrolase activity and DNA transposition was enriched in *P. myriotylum* compared to other *Pythium* species. Compared to other *Pythium* plant pathogen genomes, *P. myriotylum* has two to three times higher number of plant cell wall degrading enzymes. Many of the virulence genes were found in tandem arrays. NOTUNG analysis supported the presence of tandem arrays by gene duplication. Transcriptome of *P. myriotylum* infecting ginger, showed that proteases and plant cell wall degrading enzymes were upregulated. Genes identified in the comparative and transcriptomic analysis were shown to be related to pathogenicity (necrosis, and polysaccharide degradation) in infection and enzyme activity assays.

Correct statistical analyses were performed and shown in the figures or methods. However, these statistical analyses can also be mentioned in the main text

Results:

Line 132, please, state that a student's t-test was done and p-value for the disease index assay.

Line 152, mention that 145 single copy proteins were used for the phylogenomic analysis, and type of approach (Max Likelihood)

Line 184, indicate that Bonferroni correction applied to GO enrichment analysis p-values

Fig 5A, need higher resolution to see PCA plot

Line 364, indicate the statistical test (ANOVA)

Line 677-685, For differential expression analyses, it is better to use normalization methods of DESeq2 or EdgeR, median of ratios or TMM, rather than FPKM. The authors use DESeq2 which takes raw counts and normalizes the data so I am curious as to how they were able to work with FPKM.

Comments to Editors:

Daly and coworkers have shown that *P. myriotylum* has an expanded repertoire of virulence related genes from tandem duplication. They did an extensive amount of genomic and transcriptomic analyses to support their findings. In addition, their sequencing results also identified important genes related to pathogenicity that they were able to confirm using disease and enzyme activity assays. Not only do the authors data support their conclusions, the manuscript is well-written. Only minor comments on citing statistical tests in the main text and normalization in the transcriptomic analysis.

Re: Spectrum02268-21 (Genome of *Pythium myriotylum* uncovers the most extensive arsenal of virulence-related genes amongst the broad-host range necrotrophic *Pythium* plant pathogens)

Reviewer #1 (Comments for the Author):

The manuscript by Daly and colleagues reports on the genome sequencing, annotation, and analysis of *Pythium myriotylum*, a broad host range necrotrophic pathogen. It also reports on an expanded gene repertoire compared with other, related pathogens, and links these expansions to the occurrence of tandem gene arrays. Furthermore, the manuscript discusses the functional analysis of some virulence-factor related gene families. The objectives of the study are clearly stated, and the generated genome and transcriptome data will be of interest to the research community. However, I have several concerns related to the data and the drawn conclusion as well as the clarity of the manuscript.

A key observation is that the number of predicted protein-coding genes is large (23k) compared with other *Pythium* species. This number is indeed very high compared with gene numbers reported for most oomycetes thus far, and thus necessitates scrutiny. It seems that at least a considerable proportion of these genes are (remnants of) transposable elements, as the author reports on the enrichment of annotations associated with TEs in their results (L200ff). It is unclear if the gene annotation was based on the repeat-masked genome assembly or on the complete assembly. At any case, the authors need to remove genes that are likely related to transposons from their predicted gene set. Only then the notion of expanded gene sets compared with other *Pythium* species could be supported, especially if it would turn out that this difference is strongly driven by differences in repeat content. Furthermore, as transposons tend to cluster in oomycetes (see e.g., Haas et. al 2001), some of the tandem 'gene' clusters might in fact be due to repeat-islands rather than genuine tandem duplications of protein-coding genes. This therefore is a major technical limitation that needs to be addressed, as it affects multiple key messages of this manuscript.

Response:

Reviewer #1 analysis and thoughtful comments are correct and highly appreciated. We did not mask the genome. We did perform an analysis of repeat content of the genome using RepeatMasker and found a relatively high repeat content. However, we decided not to use the masked genome that RepeatMasker can generate because of concerns from previous experiences with other oomycete genomes that the masking can be overly stringent, whereby the sequence of *bona fide* protein-encoding genes that are not transposons can be erroneously masked. The following article from plant genome researchers is a useful reference for both the necessity of some form of repeat masking to ensure comparable analysis of protein-encoding genes as well as the pitfalls of repeat masking where genuine protein-encoding genes can erroneously be repeat masked. Bayer, P.E., Edwards, D. & Batley, J. Bias in resistance gene prediction due to repeat masking. *Nature Plants* 4, 762–765 (2018). <https://doi.org/10.1038/s41477-018-0264-0>.

There is also an opinion amongst some researchers to not mask repetitive DNA and instead filter the gene models based on homology to transposable elements such as argued in the following article: Slotkin, R.K. The case for not masking away repetitive DNA. *Mobile DNA* 9, 15 (2018). <https://doi.org/10.1186/s13100-018-0120-9>

However, we also had a misunderstanding amongst co-authors with different expertise about how repetitive sequences from our genome were treated. We did not mask for repetitive sequences, although we wrote in error that the genome was masked in the Materials and Methods, and this statement has been corrected in the revised manuscript (Line 703 in the marked-up manuscript). When looking at Pfam and InterProScan related protein annotations for all proteins (that were included in a supporting file), we could see that repeat-related proteins (e.g., transposons) could not account for the increase in total gene number compared to other *Pythium* species, as well as that repeat-related proteins were not fused to the virulence factor related genes of interest that were expanded in number compared to other *Pythium* spp. (e.g., NLPs, cutinase/AXEs, PCWDEs).

There are some fusions of transposon genes with non-transposon genes, such as the aspartyl proteases, which stand out as a category where both transposon and peptidase domains were present within the same protein. In our revised manuscript, we have excluded proteins that contained domains found in genuine-protein encoding genes if they also contained transposon-related domains. In practice, this is similar to a protein-based repeat masking strategy.

We added the following to the M&Ms section, “Gene models representing putative transposable elements were excluded if they shared significant sequence similarity (BLASTp e-value cutoff of 1e-10) to the TransposonPSI library (<http://transposonpsi.sourceforge.net/>).”

Below, Responses_Table 1 lists a comparison of the number of *P. myriotylum* genes from different “masking” approaches to removing repeat-related sequences. Different approaches to repetitive sequences have the potential to bias the comparisons made between species (e.g. for the *P. splendens* genome (Reghu et al., 2020), it appears that only 40% of the repetitive sequences have been masked as it is stated “A combination of de novo and homology-based approaches was used to mask 40% of the repetitive sequences in the genome.”). We did not want to re-annotate the genomes of the other *Pythium* species because it makes comparisons with the previous literature on these species more difficult. So there is a small level of uncertainty, but we think it is highly unlikely that there are masked regions of the other *Pythium* genomes containing orthologs of most of the extra copies of the *P. myriotylum* virulence-factor genes or independent gene expansions in parts of these genomes that have been repeat masked, i.e., it will not invalidate our claim that *P. myriotylum* has the most extensive arsenal of virulence-related genes amongst *Pythium* plant pathogens.

The use of RepeatModeller+RepeatMasker was considered too stringent because it led to the masking of some virulence factor-related genes such as elicitor-like proteins (ELs), and putative proteases that had substantial levels of expression in the RNAseq dataset. The use of RepeatModeller+RepeatMasker did not fit well with our aim of identifying potential virulence factors in the *P. myriotylum* genome. Therefore, we chose to use the TransposonPSI library to filter-out gene models representing putative transposable elements.

Annotation	Masking approach	Gene Models after “masking”
Original annotation	No masking	22,657
Original annotation	TransposonPSI hits (1e ⁻¹⁰)	19,878
After masking	Red masker	18,783
After masking	RepeatModeller+RepeatMasker	17,538

Responses Table 1. Comparison of gene content from different masking approaches. RED (Girgis, H.Z. Red: an intelligent, rapid, accurate tool for detecting repeats de-novo on the genomic scale. *BMC Bioinformatics* 16, 227 (2015)).

RepeatModeller+RepeatMasker was the software used in our manuscript to identify repeats, but the genome was not masked using this software, and instead, genes that matched to transposons were removed from the original gene annotation.

The removal of the transposons does change several of the results (several are listed below) but it does not change the overall message that our analysis of the genome of *Pythium myriotylum* uncovered the most extensive arsenal of virulence-related genes amongst the broad-host range necrotrophic *Pythium* plant pathogens.

1. In the GO term enrichment analysis, the GO term for “DNA integration” is no longer enriched.
2. There are also changes in other GO terms, such as the enrichment of “GO:0016787 hydrolase activity” in the comparisons with the plant pathogens presumably related to changes in ratio related to the reduced number of total *P. myriotylum* genes.
3. The proportion of the secreted proteins in *P. myriotylum* is higher than previously as only seven of the transposon-related 2,799 genes that were removed had a signal peptide (and no transmembrane domain) annotation, and the total number of *P. myriotylum* genes (19,878) is now ~12% lower than previously.

With regard to the question of “repeat islands”. Yes, some of the “tandem duplications” we listed are repeat islands. In the revised manuscript, we removed the transposon annotated genes from the gene set so the tandem duplications will no longer contain these “repeat islands”.

We also repeated our bioinformatic analysis of the RNAseq data to exclude the counting of expression of gene models of transposon-related genes as in most cases where gene expression datasets are analysed, transposon-related genes are excluded, and their inclusion may also lead to biases in analysis of differentially expressed genes such as from GO enrichment.

One of the main deliverables to the research community is the genome assembly of *P. myriotylum* strain SWQ7. However, the manuscript reports very little details on the amount and quality of the data generated and of the assembly itself. For instance, the manuscript should include an indication of the yield, read N50, as well as (supplementary) figures on the k-mer distributions etc. to assess genome sizes. Given the complex nature of oomycete genomes, a k-mer spectrum as reported by KAT would be insightful, as it also reports on k-mers missing from the final assembly as well as k-mers present in multiple copies. As the genome is much bigger, details and an in-depth discussion on the number of duplicated BUSCOs (and later in the manuscript on duplicated genes in general) is essential. Lastly, as the assembly is reported as haploid, the authors need to better explain their strategy to move

from a partial diploid assembly to a haploid representation, including how many contigs were removed/purged etc. All these essential information is lacking to judge the quality of the genome assembly, which is the basis of all the reported analysis.

Response: We have added additional information to the manuscript to describe the genome assembly in more detail.

Firstly, we added details on the yield from sequencing and read N50 in Line 151 in the marked-up manuscript: “Approximately 7 Gb of Illumina 150 bp paired-end reads were generated and 17 Gb of PacBio data, with a subread N50 length of 21 Kb”.

Reviewer #1 suggested the use of KAT software and we have used a similar more recently developed software tool called Merqury, as well as including a supporting figure from GenomeScope2 k-mer analysis. The GenomeScope tool was used in our original submission but we did not include a figure with the data from this analysis.

We have added a supplementary figure showing the GenomeScope k-mer count histogram that was used to estimate the genome size and % heterozygosity. We have also added a supplementary figure showing the Merqury k-mer spectrum of the initial raw assembly versus the final assembly.

The estimated genome size from GenomeScope analysis of k-mer counting was 67.5 Mb. The initial (raw) Canu assembly was significantly larger than expected at 154 Mb. 87.6% of BUSCO genes were duplicated and significant haplotypic duplication was apparent from the k-mer spectrum plots from Merqury. This is common for long-read assemblies of heterozygous species whereby divergent haplotypes are assembled into different contigs (haplotigs), instead of being collapsed, leading to false duplications. Following removal of false/haplotypic duplications using `purge_dups` and removal of contigs originating from bacterial contamination, the assembly size was reduced to 69.2 Mb which is very close to the estimated size. The number of duplicated BUSCO genes was reduced from 87.6% to 3.4%.

We do not include estimates of genome completeness based on missing/present k-mers as this is a misleading statistic due mainly to high heterozygosity and to a lesser extent bacterial contamination (101 contigs totalling 4.35 Mb). K-mer completeness from Merqury of the final assembly was only 79%. Missing k-mers can be explained mainly by those that are unique to the alternative haplotype (the assembler that we used (Canu) does not output haplotigs) and also removed bacterial contamination. We show from BUSCO completeness (95.7%), RNA-Seq mapping rates (up to 96%) and k-mer spectra that the assembly is highly complete. Furthermore, the assembly size of 69 Mb is close to the estimated genome size of 67.5 Mb from GenomeScope analysis of k-mers. We have also included the estimated consensus quality (QV) score of the assembly, which is Q39, indicating high base accuracy.

The manuscript is not easy to follow and read and much essential information is reported in the M&M (e.g., on the genome assembly) or in the figure legends. More importantly, the motivation of specific experiments (e.g., the authors' hypothesis) and the explanation of the experiments, its results, and the authors conclusion are lacking throughout the manuscript. This, unfortunately, makes the overall manuscript very hard to comprehend. For instance, it

remains unclear why the authors focused on the esterase and NLPs for their functional assays and what their expectations have been. If they hypothesized that the tested esterase could contribute to virulence and/or pathogenicity as hinted at in the results and discussion, then the performed experiments are not sufficient to address this notion. Knock-down and/or -out experiments and pathogenicity assays would need to demonstrate a contribution to virulence, which are very challenging in oomycetes. Similarly, it is unclear why the manuscript reports on growth profiles on different carbon sources, and it is anyway questionable if the limited carbon sources tested would be suitable to uncover correlations in growth and the presence and/or expansion of CAZymes. Lastly, one of the key limitations of the study, the impact of the genome assembly technology and gene annotation on the reported results is only addressed very late in the discussion. While I agree with the authors that the expansion of many gene families will likely not be impacted by erroneous gene annotation, the impact of the inclusion of transposable elements on the overall gene number and on the occurrence of tandem gene arrays needs to be addressed.

Response:

We tried to improve the readability of the manuscript. In the submitted version, it was written with a word limit in mind and thus important information was not repeated or elaborated.

In hindsight, a combined Results and Discussion section would have been clearer for the latter three sections to directly integrate the paragraphs in the Discussion related to these results sections. The structure of the manuscript starts from genome assembly & annotation, to gene expression during infection, and characterisation of activities of members of selected gene families that were expanded in *P. myriotylum* and that were upregulated during infection.

The experimental results are limited, and we plan in our future manuscripts to delve in more detail about NLPs and esterases. Yes, ultimately, we would like to test if the NLPs and esterases that we analyse contribute to virulence of *P. myriotylum* towards tobacco and ginger, but our *P. myriotylum* genetic transformation system is not optimized yet. And ideally probe the activities of all members of the gene families to understand the range of functions within these expanded gene families.

Unfortunately, at present for the putative cutinases using the data in the manuscript, we cannot formulate a very good hypothesis and so instead we state in the results section “Cutinases from oomycetes had not been heterologous produced previously for characterization of their enzymatic activities, and so to investigate the enzymatic activities of these putative cutinase/AXEs...”.

Examples of improvements in readability:

The section on “High quality genome assembly of *P. myriotylum*” has been expanded to include information in the figure, tables, and M&Ms, as well as explaining the step-by-step approach to obtaining the *P. myriotylum* assembly.

The results section on NLPs has been expanded to introduce NLPs better and explain the phenotype of the positive control. We added extra information to introduce the NLPs in the Results section rather than in the Introduction. We added a line to describe our limited hypothesis that was testable based on the experiments we performed (...

given the necrotrophic lifestyle of *P. myriotylum*, along with the presence of the residues and motifs for necrotic activity, our hypothesis was that at least some of the *P. myriotylum* NLPs would cause necrosis in *N. benthamiana* leaves).

We modified the start of the section on the growth profiling to present our hypothesis (... that there would be better growth of *P. myriotylum* on plant biomass substrates for which the *P. myriotylum* genome encoded more enzymes for their degradation). We modified the supporting figure on the growth profiling to better understand the trends by grouping the carbon sources instead of the isolates. Yes, there is a limit to how much can be inferred from the growth profiling experiments but we think it is useful for the points we want to make especially that a genome that encodes many more endo-xylanases does not grow better on xylan than other species whose genomes encode fewer endo-xylanases.

Yes, the genome assembly technology is a difference between the *P. myriotylum* genome assembly and most of the other *Pythium* genomes but we think these technical differences will not explain the expansion of many gene families. Reviewer #1 is correct that our erroneous inclusion of transposable elements will increase the total gene number and we have now revised the datasets and manuscripts to take this into account as described in the response to the previous section.

Additional concerns, comments, and suggestions:

L47: 'plant-virulent' - it is obvious what the manuscript attempts to say, but it sounds strange to refer to a strain isolated from diseased ginger rhizomes as being plant-virulent. I would suggest omitting the phrase.

Response: "plant virulent" has been deleted.

L59: Throughout the manuscript it remains unclear how the observation of gene expansions in *P. myriotylum* will lead to knowledge-driven disease management strategies. The authors need to clarify this in the discussion or reconsider the conclusion of their work.

Response: It is more of a discussion point than a conclusion that our genome sequencing and subsequent analysis will lead to knowledge-driven disease management strategies. Our conclusions are listed in the last paragraph of the results.

We added one example to the discussion:

The identification of *P. myriotylum* NLPs with necrosis-inducing activity suggests that these NLPs could be promising targets, as recently it was shown that NLPs from other plant pathogens are targets for inhibition by novel phytopharmaceutical agents [XX].

Pirc, K., V. et al., (2021). "Nep1-like proteins as a target for plant pathogen control." PLoS Pathog 17(4): e1009477.

L66: 'Pythiums' ◇ 'Pythium species'

Response: 'Pythiums' was changed to "Pythium genus"

L84: 'Here we ...' I would suggest moving this sentence towards the end of the introduction and connected with the rest of the summary of this work.

Response: This was moved as the reviewer suggested.

L129: The period is too complex and difficult to follow.

Response: This sentence was split into two sentences.

“Our previous study indicated that two *P. myriotylum* isolates, SWQ7 and SL2, had a relatively high and low virulence towards ginger [12]. Here the two *P. myriotylum* isolates were analysed in a pot trial with more replicate plants, and to measure differences at more time points in the virulence of the isolates (Fig. 1).”

L138: The statement is very relevant for the manuscripts narrative and thus not be placed in brackets.

Response: The brackets were deleted.

L142: More details should be reported at this stage on the sequence technology and genome assembly strategy.

Response: Yes, this section was expanded and was described in the response to Reviewer #1 previous comment.

L147: What is meant by 'The final...'? The sentence seems grammatically not correct and necessitates further explanation. For instance, what do the authors mean by 'haptotypic duplications' and why the assembly from long-read was considered to be a diploid assembly.

Response: 'haptotypic ...' was a typo and should have been “haplotypic...”. Reviewer #1 correctly pointed out that we did not describe our assembly in sufficient detail. We have described our additions to the manuscript in the response to Reviewer 1’s previous comment.

L151: This sentence is grammatically not correct

Response: This sentence has been simplified.

“Also, the predicted assembly size of the *P. myriotylum* SWQ7 isolate based on k-mer analysis of Illumina short reads was similar to another *P. myriotylum* isolate SL2.”

L154: The authors need to provide more detail on their analysis rather than merely referring to it as 'phylogenetic analysis'. Why was this analysis performed? On which marker was this tree based? Some of this information is obviously in the M&M but the manuscript becomes incomprehensible without the essential information.

Response: The text has been modified to describe the reason for performing the analysis and provide details of the methods.

“A phylogenomic analysis was carried out to confirm the evolutionary relationship between *P. myriotylum* and other sequenced *Pythium* species. Two *Phytophthora* species were included as outgroups. Maximum-likelihood analysis was performed on a concatenated supermatrix alignment of 145 BUSCO proteins that were present as single-copy in at least 11 of the 12 species.”

L197: Could the statement 'which are often secreted' please be quantified by providing the number of secreted proteins?

Response: The below was added. Note that GO:0004175 (endopeptidase activity) was replaced with GO:0006508 (proteolysis) because this represented a broader range of protein hydrolases.

“...which are often secreted (99 and 382 of the proteins annotated with GO:0004553 and GO:0006508, respectively, were predicted to be secreted), ...”

L224. This sentence is grammatically not correct

Response: The sentence was split into two sentences, and the word “with” was added near the start of the first sentence.

“Correlating well with the presence of the largest number of xylanases in *P. myriotylum*, it was the only *Pythium* species containing genes encoding GH62 family members in the CAZy analysis. Furthermore, in the ortholog analysis, all four of the GH62 family members from *P. myriotylum* are found in the MCL_06066 group which is unique to *P. myriotylum* (Table S2).”

L254: 'above a threshold' \diamond this is very vague and should be clarified

Response: We have repeated what was written in the M&Ms in parenthesis “(an E value of less than $1e^{-10}$ and highest-scoring pair length greater than half the length of the shortest sequence)”.

L262: 'While this trend...' I am unfortunately not able to follow the authors' argument. Could they please rephrase this sentence to clarify their argument?

Response: Apologies as this sentence was poorly constructed and has been modified as follows:

“This trend whereby *P. myriotylum* has the highest number of tandem arrays and genes within tandems is not because *P. myriotylum* has more genes than most of the other species because when normalized for total gene number, the percentage of genes in *P. myriotylum* found within tandem arrays (27%) was also higher than all of the other *Pythium* species”

L278: The definition of tree reconciliation is not entirely correct. NOTUNG, as many other tree reconciliation algorithms, aims to explain discordance between species and gene trees and not, as mentioned by the authors, whether gene content differences are due to gains. Based on the expansion of NLPs, and many other gene families, in *P. myriotylum*, it is to be expected that these are due to duplication. However, the interesting question not addressed by the authors is if these duplications are *P. myriotylum* specific or are ancestral and gene copies have been lost in sister species, which can be derived from the NOTUNG analyses. Furthermore, it would be relevant (see also comment below) if recent duplicates are organized in tandem arrays or not.

Response: The text has been edited to modify the definition of tree reconciliation.

The NOTUNG analysis did also output gene loss events, as shown in Responses_Figure 1 below. We did not include the image with the gene loss events in the submitted manuscript as we considered the gene loss events less reliable due to the limited number of used genomes that were relatively closely related to *P. myriotylum*. The gene loss events were far fewer than the duplication events in *P. myriotylum*, indicating that the duplication events are unlikely to be ancestral. Some of the recent duplicates are all

organised in a single tandem, such as Pm_g13401 to Pm_g13406. It is not the case with the other two major duplications in *P. myriotylum* where the NLPs here are not all part of the same tandem array but are dispersed in several tandem arrays in different parts of the genome.

Responses_Figure 1. NOTUNG analysis showing putative gene loss events indicated with grey-colored text.

L292: It is unclear from the text what analysis has been performed.

Response: The following has been added “...in a principal component analysis of the transcriptomic data...”

L317, L319: 'Between 10 to 16...' 'Despite the large number...' how many cellulases or xylanases are encoded in the genome? To aid interpretation, the manuscript should report these numbers here too.

Response: The total number of cellulase and xylanases in the genome has been added here.

L332: 'relatively low levels' this is very vague. What is relatively low? What is high? The authors need to offer context.

Response: We apologise for the lack of clarity. We define expression as low (<10 FPKM) moderate (10 < FPKM < 100) or high (> 100 FPKM) based on FPKM levels. For those genes with low expression (<10 FPKM), we do not consider these as upregulated even if there is a statistically significant difference. The particular putative CRN effector that Reviewer #1 asked about here was not considered upregulated in the re-analyzed

transcriptome dataset as the FPKM value during infection was <10 FPKM.

L346: Remove the () around the explanatory statement as it is relevant to understand the experiment and its results.

Response: These were removed.

L371: Three of the AXE genes were upregulated and, based on the gene code, seem to be localized in a single locus. What is the nucleotide similarity between these genes and could overall similarity between genes in these tandem arrays impact the gene expression analysis, e.g., due to multi-mapped RNAseq reads?

Response:

In the case of the three CUT/AXE genes – Pm_CUT/AXE-4 (Pm_g15824.t1), Pm_CUT/AXE-7 (Pm_g15827.t1) and Pm_CUT/AXE-8 (Pm_g15828.t1) – these three genes are sufficiently different to distinguish these from each other. The maximum nucleotide identity is 88% (Responses_Table 2). The 150 bp paired-end reads will span regions that are different in each of the three CUT/AXE genes. Yes, these three genes are localized to the same locus.

	Pm_CUT/AXE-4 (Pm_g15824.t1)	Pm_CUT/AXE-7 (Pm_g15827.t1)	Pm_CUT/AXE-8 (Pm_g15828.t1)
Pm_CUT/AXE-4(Pm_g15824.t1)	100	88.12	73.8
Pm_CUT/AXE-7(Pm_g15827.t1)	88.12	100	74.81
Pm_CUT/AXE-8(Pm_g15828.t1)	73.8	74.81	100

Responses_Table 2. Nucleotide percentage identity matrix for the three CUT/AXE coding sequences (CDS) that were upregulated during infection of ginger demonstrating that the sequences are sufficiently different for RNAseq to detect the expression of each gene. Data generated from <https://www.ebi.ac.uk/Tools/msa/mafft/>.

Responses_Figure 2. Sequence alignment for the three CUT/AXE coding sequences (CDS) that were upregulated during infection of ginger demonstrating that the sequences are sufficiently different for RNAseq to detect the expression of each gene.

With regard to the more general question from Reviewer #1 about the overall similarity between genes in these tandem arrays and the impact on the gene expression analysis. Multi-mapping reads are not likely to be a substantial problem in the analysis of expression of genes in tandem arrays.

There was 806/19,878 (~4%) of the ".t1" CDS nucleotide sequences that were 100% identical to at least one other ".t1" CDS sequence. 490/806 of these were found in tandem array (in total 5,099 genes are found in tandem arrays). However, is it still possible to detect specific expression for these identical genes because the 5' and 3' UTRs can anchor the reads. The UTRs were not annotated by the gene annotation method (most tools perform poorly at annotating UTRs) and therefore, the UTRs were not used along with the CDS sequences in the above calculation. The ".t1" CDS sequences that are identical to at least one other ".t1" CDS sequence are listed in the supporting Excel with the RNAseq data to flag to the reader that low or absence of expression may be a false negative.

Counting multi-mapping reads does not appear reliable or possible using the pipeline we used for RNAseq analysis (HISAT2 aligner and HTSeq-count). Selecting the "non-unique" option in HTSeq-count can lead to artifacts such as expression values for genes with limited coverage of the entire CDS.

On average, ~5% of the RNAseq reads were classed as multi-mapping by HISAT2 and >85% aligned exactly one time to our genome assembly.

Also, in the revised supporting file of the tandem duplications, we included a column of the average percentage protein identity of the tandem arrays. Here only 95/1,666 arrays tandem arrays have 100% average identity, so this also supports that multi-mapping reads are unlikely to be a substantial problem in the analysis of expression of genes in tandem arrays.

L393: The manuscript needs to properly introduce NLPs as well as the differences between type I/II, toxic/non-toxic as well as the expected phenotypes based on the positive control INF1.

Response: Several sentences have been added to the start of the results paragraph on NLPs to better introduce NLPs (part of this is taken from the Discussion), and information on type and toxicity is added as well as an explanation of the phenotype caused by the INF-expressing *Agrobacterium* strain.

Note that there was another change to this section whereby *Pm_NLP-14* was described as moderately induced during infection by one of the *P. myriotylum* isolates. In the repeat analysis of the transcriptome datasets (after exclusion of the transposon-related genes), the expression of *Pm_NLP-14* was slightly below an FPKM of 10 and therefore, the gene was not considered significantly induced.

L427: This statement necessitates an appropriate reference.

Response: The following recent review about gene expression changes contributing to host immunity was added:

Chen, H., S. Raffaele and S. Dong (2021). "Silent control: microbial plant pathogens evade host immunity without coding sequence changes." *FEMS Microbiology Reviews*

45(4).

L438: 'The *P. myriotylum* infection strategy...' This statement is highly speculative, and it is not clear to me how this conclusion is supported by the presented data.

Response: This sentence was deleted.

L446: 'A lack of ...' this statement could be further supported by checking the genome sequence for the presence of the corresponding enzymes.

Response: There are putative enzymes that could be part of the xylose catabolic pathway in *P. myriotylum* based on Pfam annotations using fungal enzymes as an example from *Aspergillus niger* (Responses_Table 3). However, there are many *P. myriotylum* proteins with these Pfam annotations, and there is a large evolutionary distance between oomycetes and species where these catabolic enzymes and pathways have been characterized. Therefore, it requires some functional analysis, e.g., characterization of heterologous enzymes and/or gene deletion to identify the enzymes involved in xylose catabolism in oomycetes. We think this speculation is too weak to add to the Discussion.

Pfam domain(s)	EC number	activities/functions
PF00106	1.1.1.10	L-xylulose reductase
PF00106	1.1.1.10	L-xylo-3-hexulose reductase
PF00107; PF08240	1.1.1.12	L-arabinitol 4-dehydrogenase
PF00107; PF08240	1.1.1.9	xylitol dehydrogenase
PF08240	1.1.1.9	D-xylulose reductase
PF02782	2.7.1.17	D-xylulose kinase

Responses_Table 3. **List of Pfam domains related to catabolism of xylose based on *Aspergillus niger* genome annotation where several enzymes have been characterized: Aguilar-Pontes MV, et al. The gold-standard genome of *Aspergillus niger* NRRL 3 enables a detailed view of the diversity of sugar catabolism in fungi. *Stud Mycol.* 2018;91:61-78. doi:10.1016/j.simyco.2018.10.001**

L464: Another potential explanation for the occurrence of an expanded set of NLPs is the potential role of NLPs in niche establishment as NLPs have been commonly found to also occur in saprophytes.

Response: Yes, that is a useful point, and a sentence has been added about this point on Line 587 in the marked-up manuscript. It is also possible some of these saprotrophic fungi could be plant pathogenic of particular undiscovered hosts.

L575: Which PacBio long-read sequencing platform was used for this study?

Response: The SWQ7 isolate was sequenced using PacBio continuous long-read sequencing (Sequel system instrument), and this information has been added to the M&Ms on Line 684 in the marked-up manuscript.

L586: How many corrections to the genome assembly have been performed by Pilon for each iteration? How many changes or uncertainties were retained after the 2nd iteration?

Response: Unfortunately, we do not have this level of detail retained from using Pilon to polishing the genome assembly. From the Merqury analysis, we have also included the

estimated consensus quality (QV) score of the assembly, which is Q39, indicating high base accuracy.

L692: The authors applied a cutoff of FPKM > 10 to differentiate between expressed and non-expressed genes. How has this cutoff been chosen, and how many genes are not expressed under this cutoff? Furthermore, on line 326 the manuscript mentions that three NLPs are highly expressed based on an FPKM of 100. If genes < 10 are considered non-expressed, then a FPKM of 100 does not seem to be particularly high. How does this relate to the expression levels of other genes in the genome?

Response: Yes, a cut-off of FPKM > 10 was used to exclude genes below this level that were considered lowly expressed. Genes of FPKM < 10 can still be considered expressed (i.e., they are not “non-expressed”) but their contribution to the functional changes (i.e., the infection of ginger leaves) is likely to be less meaningful than genes whose FPKM > 10. This cut-off was partly chosen because there were several thousand genes up and downregulated using the FPKM > 10 cut-off, and it was considered more biologically relevant to focus on the analysis of these genes (FPKM > 10), such as the for the lists of differentially expressed genes for the GO enrichment analysis (the GO enrichment analysis does not take the level of expression into account). Approximately ~1,500 genes were excluded because their FPKM < 10 even though they meet the fold change and P value criteria. The below table lists the total number of these genes.

In the supporting file, all the genes are included, and the reader can select alternative cut-offs if they so wish (e.g., an FPKM >1).

Category	Cut-off FPKM > 10	Without an FPKM cut-off
SWQ7 ↑ infection	2,110	3,134
SWQ7 ↓ infection	2,302	4,108
SL2 ↑ infection	2,513	3,874
SL2 ↓ infection	2,995	4,440

Responses_Table 4. Comparison of the number of DE genes with and without a cut-off of FPKM > 10.

With regard to the question about the expression levels (e.g., FPKM > 10 and FPKM > 100) and how they are related to the expression of other genes, a quantile analysis can show this (Responses_Table 5) e.g., the quantile of 92% is where 92% of the genes have an FPKM value lower than ~100 FPKM so therefore, ~8% of the genes have an FPKM value > ~ 100 FPKM, and ~ 40% of genes will have an FPKM value > ~10. These of course are average values, and which genes belong to which quartile will change depending on control or infection condition.

sample	mean	sd	IQR	0%	the quantile of XX% is where XX% of the genes have an FPKM value lower than indicated											
					10%	20%	30%	40%	50%	60%	70%	80%	90%	92%	95%	100%
in_SL2_1	57.1	386.3	24.1	0.0	0.0	0.0	0.3	1.7	4.5	9.6	18.1	33.1	74.8	95.6	155.2	11165.5
in_SL2_2	58.5	409.2	23.8	0.0	0.0	0.0	0.4	1.6	4.3	9.3	17.7	32.1	73.2	93.3	155.4	12200.8
in_SL2_3	58.6	413.7	23.6	0.0	0.0	0.0	0.3	1.5	4.2	9.1	17.6	32.0	73.5	93.6	156.4	11950.1
in_SWQ7_1	58.9	406.6	24.4	0.0	0.0	0.0	0.5	1.9	4.7	10.0	18.3	33.5	75.7	95.6	152.9	12240.8

sample	mean	sd	IQR	the quantile of XX% is where XX% of the genes have an FPKM value lower than indicated												
				0%	10%	20%	30%	40%	50%	60%	70%	80%	90%	92%	95%	100%
in_SWQ7_2	58.3	403.7	25.5	0.0	0.0	0.0	0.6	2.1	5.3	10.5	19.6	35.0	76.4	96.0	150.8	13206.6
in_SWQ7_3	57.9	389.2	25.2	0.0	0.0	0.0	0.5	2.0	4.9	10.3	18.9	34.3	77.4	97.8	155.9	12041.2
m_SL2_1	49.7	311.7	15.3	0.0	0.0	0.0	0.2	0.8	2.2	4.9	10.5	23.0	66.6	88.6	154.7	15402.7
m_SL2_2	49.5	305.3	15.8	0.0	0.0	0.0	0.2	0.9	2.3	5.2	10.9	23.7	67.0	90.3	156.4	15160.7
m_SL2_3	49.3	302.1	16.5	0.0	0.0	0.0	0.3	1.0	2.5	5.5	11.4	24.7	68.7	92.4	157.5	14774.3
m_SWQ7_1	48.9	233.2	26.0	0.0	0.0	0.1	1.0	2.9	6.1	11.0	19.7	36.3	80.7	102.7	162.2	10271.9
m_SWQ7_2	48.5	221.0	27.2	0.0	0.0	0.1	1.0	3.1	6.6	11.9	20.9	37.5	81.9	104.6	162.8	8558.4
m_SWQ7_3	48.5	220.6	27.8	0.0	0.0	0.1	1.0	3.1	6.7	12.0	21.3	38.2	83.2	106.1	164.4	9090.9
average	53.6	333.5	22.9	0.0	0.0	0.0	0.5	1.9	4.5	9.1	17.1	32.0	74.9	96.4	157.1	12172.0

Responses_Table 5. Summary of the FPKM expression data quantiles for each sample and the average values of all the samples.

Table 1: It is unclear how the species in Table 1 are ordered. If there is no specific rationale in the order, I would suggest considering to order based on the phylogeny show in Figure 2.

Response: Apologies, the species were ordered based on whether their host was a plant, fungal/oomycete host, or animal host. This has now been written on the Table. It is in alphabetical order within these host categories except for *P. myriotylum*. This follows the same naming categorization used in the phylogenetic tree in Fig. 2 in the manuscript.

Table 2: It is unclear why the sum of all proteins per category would be informative as it depends on the number of species considered. For clarity, the table should omit this column.

Response: The column was deleted.

Figure 1: Panel B needs to report the individual datapoints underlying the barplots to enable readers to assess the spread of the data. Furthermore, it would be useful if they figure could also show representative figures for different disease index to judge how DI relates to symptoms.

Response: The figure was modified to show the individual datapoints showing the spread of data as well as highlighting different levels of disease symptoms.

Figure 2: The authors could consider integrating some data from table 1 (e.g., genome size, repeat content, and/or gene content) into this figure.

Response: We prefer to keep the data from Table 1 together instead of transferring them to the figure.

Figure 3: To ease interpretation of the figure, panel A should report the ratio or the fold-change rather than the individual numbers.

Response: We prefer to use the individual numbers in panel A because they also display the number of proteins annotated with the GO term, which would be absent if a ratio or fold change was used.

Figure 4: To further support the frequent occurrence of tandem arrays in *P. myriotylum*, the authors should consider showing few example regions. Panel c should report the ratios, fold-changes, and p-values of the overrepresented categories.

Response: This Figure was modified with these changes.

Figure 5: It is counter intuitive that the dimensionality reduction of the transcriptomes of each isolate is reversed in the same panel. Furthermore, the vertical line in the panel for SL2 is not explained and unclear. Panel D should relate the number of up-regulated genes to the total number of genes in each category and each strain.

Response: The dimensionality reduction explained by the y-axis explains less than 10% of the total reduction whereas the x-axis explains about 80%. The trend in the dimensionality reduction that explains about 80% is the same for both isolates. I think the reversal of the dimensionality reduction may be counter-intuitive, it is not a problem for the points about the dataset that the PCA is used to make.

The number of up-regulated genes was related to the total number of genes in each category by adding the total number of genes in parenthesis on the y-axis.

Figure 6: The color key for the heatmap is not sufficiently labeled. One of the replicates of the control for SL2 behaves different than the other replicates. To which extent does this impact the differential expression analysis (as mentioned in L300)? To obtain robust gene expression, the authors should consider repeating the expression experiment. The labels on top of the inoculation areas need to be moved to the side to aid visual inspection of the necrotic areas. Furthermore, the necrotic areas are hard to see and judge. The authors should complement the visual representation of the analysis with quantification of necrotic areas, for instance by using red light imaging (Villanueva et al. 2021). As mentioned for Fig. 4 the authors should consider showing a genomic locus of recently expanded NLPs to demonstrate how these evolved in tandem arrays, see e.g., Dong et al. 2012; Fig. 6.

Response: With regard to the greater variability in the control replicates for the SL2 isolates, there was an error with the SL2 isolate paired read files whereby the "R2" of the pair from one of the infection samples instead of the control sample was used. HISAT2 flagged an error about the wrong pairs but it was not noticed at the time because the software still outputted a bam file, and the clustering results from the PCA of the replicates were sensible (it seems that the expression from this "replicate" was ~ 5% from the "R2" pair from treatment and ~95% from the "R1" pair from control. In the revised manuscript, the correct "R2" pair sample was used when the transcript analysis was repeated to generate the transcriptome dataset without transposon genes. The sentence about greater variability in the replicates in the control was deleted. We apologise for this error. It does not change the overall trends or conclusions of our transcriptome results.

Reviewer #1 made a comment recommending the repeat of the transcriptome experiment to ensure robust results but this was based on the variability of the control replicates for the SL2 isolate, and so this issue is no longer of concern. Furthermore, the use of two different isolates partly functions as pseudo-replicates as similar expression trends are seen during infection from both isolates. The use of two isolates of *P. myriotylum* gives more confidence in the reproducibility of the expression data.

With regard to the images of the necrotic lesions, the labels were moved to the side of the lesion. The quantification of necrotic areas by using red light imaging (Villanueva et al. 2021) is a very useful suggestion for the future but we unfortunately did not perform such quantification in our experiments. The purpose of showing the images of the necrotic lesions is to demonstrate an example leaf for each of the six NLPs. The "++"

and the “+” in the necrosis column in the table is referring to the relative timing of the necrosis compared to the INF1 positive control (this is described in the results section but not on the figure itself). The “++” and the “+” is not referring to differences in the necrotic lesions in the images because the images of the leaves were taken at the end of the experiment and the earlier differences in necrosis compared to the INF1 control are no longer apparent.

Figure S2. A substantial number of proteins have both signal peptides as well as TMHMM. This suggests that either the protein is not a secreted ELL or that the SignalP and/or TMHMM prediction is incorrect. Furthermore, some ELL do not have a signal peptide. What does this suggest in terms of their biological function and/or gene annotation?

Response: 10/100 of the ELLs have both a signal peptide and a TMHMM domain. The presence of both a signal peptide and TMHMM annotation does not necessarily indicate an annotation error. The presence of a signal peptide is also required for secretion into a membrane for Type I membrane proteins. There is previous research that shows ELLs can contain transmembrane domains: Qutob, D., Huitema, E., Gijzen, M. and Kamoun, S. (2003), Variation in structure and activity among elicitors from *Phytophthora sojae*. *Molecular Plant Pathology*, 4: 119-124. <https://doi.org/10.1046/j.1364-3703.2003.00158.x>. For ELLs that do not have a signal peptide, it is possible that they are secreted by the non-conventional secretion pathway. As ELLs can function as a PAMP, there can be selective pressure for loss of function of particular ELLs to evade recognition by the plant host. Concerning the annotation, there can be a level of uncertainty in the gene model predictions, especially for ELLs without RNAseq data to aid the gene model prediction.

Reviewer #2 (Comments for the Author):

Daly and co-workers have isolated the highly virulent SWQY strain of *Pythium myriotylum* from spice ginger. In a GO enrichment analysis, hydrolase activity and DNA transposition was enriched in *P. myriotylum* compared to other *Pythium* species. Compared to other *Pythium* plant pathogen genomes, *P. myriotylum* has two to three times higher number of plant cell wall degrading enzymes. Many of the virulence genes were found in tandem arrays. NOTUNG analysis supported the presence of tandem arrays by gene duplication. Transcriptome of *P. myriotylum* infecting ginger, showed that proteases and plant cell wall degrading enzymes were upregulated. Genes identified in the comparative and transcriptomic analysis were shown to be related to pathogenicity (necrosis, and polysaccharide degradation) in infection and enzyme activity assays.

Correct statistical analyses were performed and shown in the figures or methods. However, these statistical analyses can also be mentioned in the main text

Response: We have added information on the statistical methods in the main text. Please note that substantial other changes have been made in response to Reviewer #1.

Results:

Line 132, please, state that a student's t-test was done and p-value for the disease index assay.

Response: This was added on Line 142 in the marked-up manuscript.

Line 152, mention that 145 single copy proteins were used for the phylogenomic analysis, and type of approach (Max Likelihood)

Response: This information has now been added starting on Line 194 in the marked-up manuscript.

Line 184, indicate that Bonferroni correction applied to GO enrichment analysis p-values

Response: This was added on Line 241 in the marked-up manuscript.

Fig 5A, need higher resolution to see PCA plot

Response: The resolution of the PCA plots were improved, as well as simplifying the labels on the plots.

Line 364, indicate the statistical test (ANOVA)

Response: From this growth profiling, a statistical test was not used here. The differences being described are primarily visual differences and broader trends.

Line 677-685, For differential expression analyses, it is better to use normalization methods of DESeq2 or EdgeR, median of ratios or TMM, rather than FPKM. The authors use DESeq2 which takes raw counts and normalizes the data so I am curious as to how they were able to work with FPKM.

Response: For the differential expression analysis, DESeq2 was indeed used, and functions as described by Reviewer #2. Separately to our DESeq2 analysis, the count data from HTseq-count was used to calculate FPKM. Because DESeq2 normalization does not correct for gene length, we used the FPKM values alongside the DESeq2 fold changes and Padj values. The fold changes from DESeq2 are not identical to those from using FPKM values, and it is noted in the M&Ms and supporting files that the DESeq2 fold changes were used.

Reviewer #3 (Comments for the Author):

This work does a great job summarizing the extensive repeat-driven diversification of the *Pythium myriotylum* genome and how it contributes to its success as a broad-range necrotrophic plant pathogen. Of particular note is the enrichment of tandem repeats which appears to be driving the substantial increase in total gene count.

It is impressive to assemble such a repeat-laden oomycete genome sequence to a scaffold N50 of 1.6 Mb and only 185 total contigs. However, some statistics such as contig number, N50, and repeat percentage should be more explicitly stated in the text (around line 161), and repeat content is only ever mentioned in the entire manuscript in the discussion on line 512. This would be a useful statistic to include in table 1 in comparison to the other sequenced *Pythium* species. Additionally, the particular PacBio technology (HiFi reads??) needs to be stated in the methods (line 575).

Response: The repeat content was mentioned in the text rather than in the table because it can be misleading to compare repeat content with the other *Pythium* species as most are from second generation genome sequencing which underestimates repeat content compared to third generation genome sequencing. Some of the parts that were in the Discussion about repeat content were moved to the Results section.

With regard to the particular PacBio sequencing technology, continuous long-read sequencing (Sequel system instrument) was used and this has been stated on Line 684 in the marked-up manuscript. In the results section on “High quality genome assembly of *P. myriotylum*”, further information on the PacBio sequencing data has also been included (“17 Gb of PacBio data, with a subread N50 length of 21 Kb”).

Information such as contig number and N50 were added to the section on “High quality genome assembly of *P. myriotylum*”.

With regard to the repeat content, yes, Reviewer #3 is correct that more information on this needs to be added to the results. We have actually deleted one paragraph from the results related to repeat content starting on Line 200 in the submitted manuscript. This paragraph was about an enrichment for genes involved in DNA transposition in *P. myriotylum* compared to the other *Pythium* species analyzed. However, we did not filter-out putative transposon genes from our genome annotation, and after these transposon genes were filtered out, there was no longer an enrichment in *P. myriotylum* for genes related to DNA transposition. Please see our detailed response to Reviewer #1 for how we have removed repeat-related sequences to avoid biasing our comparisons with the other *Pythium* species.

In relation to the analysis of tandemly duplicated genes, one statistic that would be useful is the average % identity shared between members of each array. Are they often identical in sequence, or have they substantially diverged? A related question would be: what is the inferred timeline of these duplications based on sequence divergence? Are these recent, or are they broadly similar to divergence rates of other oomycetes? Are only the NLPs diverging rapidly compared to the rest of the genes (line 483)?

Response: We did an analysis of the % identity shared between members of a tandem array. In the revised supporting file of the tandem duplications, a column was included

of the average percentage protein identity of the tandem arrays. After the filtering-out of genes matching to transposons, our re-analysis of the tandem arrays in the *P. myriotylum* genome found 1,666 tandem arrays. Here only 95/1,666 tandem arrays have 100% average identity. The average protein identity between members of tandem arrays ranges from 23% to 100%. The average of these average percentage identities is 62%, indicating that there is substantial divergence amongst members of the tandem arrays.

With regard to the inferred timeline of these duplications based on sequence divergence, we did not do this analysis. Based on the analysis of the NLPs found in tandem arrays, they appear to most likely be recent duplication events. About half of the MCL groups that the tandem arrays are members of tend to have more *P. myriotylum* members than of other species (with the exception of *P. guiyangense* which is a hybrid), and this implies that the timeline of at least some of the duplication is after the divergence of *P. myriotylum* from the other *Pythium* species. Of course, gene loss in other *Pythium* species could also explain the higher number of genes in *P. myriotylum* but this is difficult to determine from a limited number of species.

The dN/dS analysis of the NLPs indicated that they are under purifying selection and so are not diverging rapidly (the dN/dS ratio was < 1 indicating purifying selection). We did an analysis of the selective pressure using the average dN/dS ratio of members of a tandem array. Of the 1,666 tandem arrays, most (1,655) were under purifying selection (dN/dS ratio < 1), and only 11 were under diversifying selection (dN/dS ratio > 1). This data is included as an extra column in the Supporting file of the tandem repeats, and a paragraph has been added to the results section on tandem arrays.

In relation to the analysis of the PCWDEs, my specific expertise is not in the profiling of gene activity and biochemistry. However, I believe the authors have done a thorough job exploring the interesting features of the extensive repertoire of *P. myriotylum* virulence factors, especially when experimentally confirming the activity of NLP proteins.

Response: We thank Reviewer #3 for the positive comments.

June 30, 2022

Dr. Paul Daly
Jiangsu Academy of Agricultural Sciences
Institute of Plant Protection
No. 50 Zhongling Street
Nanjing 210014
China

Re: Spectrum02268-21R1 (Genome of *Pythium myriotylum* uncovers an extensive arsenal of virulence-related genes amongst the broad-host range necrotrophic *Pythium* plant pathogens)

Dear Dr. Paul Daly:

Your manuscript has been accepted, and I am forwarding it to the ASM Journals Department for publication. You will be notified when your proofs are ready to be viewed.

Sincerely,

Lindsey Burbank
Editor, Microbiology Spectrum

Journals Department
Supplemental Dataset: Accept
Supplemental Dataset: Accept
Supplemental Dataset: Accept
Supplemental Dataset: Accept
Supplemental Material: Accept
Supplemental Dataset: Accept
Supplemental Dataset: Accept
Supplemental Dataset: Accept
Supplemental Dataset: Accept